# CircuitFusion: Multimodal Circuit Representation Learning for Agile Chip Design

**Wenji Fang    Shang Liu    Jing Wang    Zhiyao Xie** [*]
The Hong Kong University of Science and Technology
{wfang838,sliudx,jwangjw}@connect.ust.hk, eezhiyao@ust.hk

## Abstract

The rapid advancements of AI rely on the support of integrated circuits (ICs). However, the growing complexity of digital ICs makes the traditional IC design process costly and time-consuming. In recent years, AI-assisted IC design methods have demonstrated great potential, but most methods are task-specific or focus solely on the circuit structure in graph format, overlooking other circuit modalities with rich functional information. In this paper, we introduce **CircuitFusion**, the first multimodal and implementation-aware circuit encoder. It encodes circuits into general representations that support different downstream circuit design tasks. To learn from circuits, we propose to fuse three circuit modalities: hardware code, structural graph, and functionality summary. More importantly, we identify four unique properties of circuits: parallel execution, functional equivalent transformation, multiple design stages, and circuit reusability. Based on these properties, we propose new strategies for both the development and application of CircuitFusion: 1) During circuit preprocessing, utilizing the parallel nature of circuits, we split each circuit into multiple sub-circuits based on sequential-element boundaries, each sub-circuit in three modalities. It enables fine-grained encoding at the sub-circuit level. 2) During CircuitFusion pre-training, we introduce three self-supervised tasks that utilize equivalent transformations both within and across modalities. We further utilize the multi-stage property of circuits to align representation with ultimate circuit implementation. 3) When applying CircuitFusion to downstream tasks, we propose a new retrieval-augmented inference method, which retrieves similar known circuits as a reference for predictions. It improves fine-tuning performance and even enables zero-shot inference. Evaluated on five different circuit design tasks, CircuitFusion consistently outperforms the state-of-the-art supervised method specifically developed for every single task, demonstrating its generalizability and ability to learn circuits' inherent properties[1].

## 1 Introduction

The fast advancements of AI require the support of hardware circuits (e.g., GPU, TPU, NPU). However, the increasing complexity of the digital integrated circuit (IC) has led to skyrocketing IC design costs, challenging traditional IC design methodologies. In recent years, AI-assisted IC design techniques have demonstrated unprecedented potential in enabling agile chip design process (Rapp et al., 2021). Existing explorations include automated chip design planning (Mirhoseini et al., 2021; Bai et al., 2023; Yu et al., 2018), early chip quality evaluation (Du et al., 2024; Fang et al., 2024a; Zheng et al., 2023; Wang et al., 2023b), automated chip design assistance (Pei et al., 2024; Liu et al., 2023b;a; Yao et al., 2024), etc. These works are mostly task-specific and trained supervisely with a small amount of labeled circuit data. In this work, we target essentially general and multi-tasking AI models for circuits by exploring customized circuit representation learning techniques.

**Learning hardware representation.** Several recent works have proposed either supervised (Yang et al., 2022; Li et al., 2022b; Shi et al., 2023; Vasudevan et al., 2021; Yang et al., 2022; Deng et al., 2024) or self-supervised contrastive methods (Wang et al., 2022; Xu et al., 2023; Fang et al.,

---

[*]Corresponding Author
[1]CircuitFusion is available at: https://github.com/hkust- zhiyao/CircuitFusion

2025) to learn circuit representations. Most works only focus on the circuit *structure* by converting circuits into a graph format and learning circuit embeddings based on the graph alone. However, *circuit* is a unique data type and inherently exhibits multimodal characteristics. A register-transfer level (RTL) circuit can be described with hardware description language (HDL) code, a graph of operators, or a natural-language summary of functionality. Generally, we view each circuit as a *structured* implementation of certain *functionality*. The graph modality highlights the structure, the functionality summary emphasizes the functionality, while the HDL code incorporates information from both aspects.

**Learning multimodal representation.** To encode information from diverse modalities, multimodal representation learning has been successfully applied for various modality fusions, such as vision-language (Radford et al., 2021; Li et al., 2021; 2022a; 2023a; Akbari et al., 2021; Bao et al., 2022), graph-language (Yin et al., 2020; Gao et al., 2020), and software-graph (Guo et al., 2022a; Zhang et al., 2024). **However, no existing work has ever exploited inherent multi-modalities of RTL circuits.** More importantly, there are significant differences between the mechanisms of circuits and other common data types during multimodal representation learning, as we will summarize later.

To this end, we present **CircuitFusion**, the first multimodal fused and implementation-aware circuit encoder to capture informative general circuit representations. Leveraging the multimodal nature of circuits, we first encode circuits separately under three modalities (i.e., HDL code, structural graph, and functionality summary) through three independent unimodal encoders. A multimodal encoder then fuses the modalities via a cross-attention mechanism. Additionally, we enrich CircuitFusion with downstream circuit implementation information using an auxiliary netlist encoder.

To learn this special multimodal circuit data, we identify four unique circuit properties (P1-4) and propose four corresponding innovative strategies (S1-4), applied to circuit preprocessing (S1), CircuitFusion pre-training (S2,3), and application of CircuitFusion in downstream tasks (S4):

**P1: Parallel execution.** Hardware circuit operates with inherent parallelism, where all combinational logic calculates simultaneously, and all sequential elements are updated at every clock cycle. **S1: Sub-circuit generation.** Exploiting this parallel mechanism, we propose to split the entire circuit into multiple sub-circuits based on sequential element boundaries. This preprocessing step allows for scalable CircuitFusion learning across the entire circuit.

**P2: Functional equivalent transformation.** The same functionality of a circuit can be implemented in various ways, as each logic expression can be equivalently transformed. Therefore, circuits with similar functionality may have entirely different structures. **S2: Semantic-structure pre-training on circuit.** Leveraging this property, we design three pre-training tasks to simultaneously capture both structural (i.e., masked graph modeling) and semantic (i.e., masked summary modeling, functional contrastive) circuit representations within and across the three modalities. The circuit data for these tasks is further augmented through equivalent transformations, generating circuits that maintain the same functionality but have entirely different structures.

**P3: Multiple design stages.** In standard digital IC design flow, the functional behavior is described with HDL codes at the RTL stage. Then the RTL circuits will be automatically synthesized into gate-level netlists, representing the circuit implementation in connected logic gates. The netlists are then transformed into the chip layout for ultimate manufacturing. This multi-stage design process involves various levels of circuit abstraction, with earlier stages (e.g., RTL) capturing more semantics and later stages (e.g., netlist) introducing more implementation details. **S3: Circuit implementation-aware alignment.** Based on this multi-stage property, we propose to align the RTL circuits with their netlist implementations within a shared latent space during pre-training. This alignment integrates implementation information into circuit representations, significantly benefiting downstream tasks that predict the circuit implementation metrics at early RTL stages.

**P4: Circuit reusability.** In realistic circuit development, circuits exhibit high reusability, with companies tending to use intellectual property (IP) blocks rather than designing circuits from scratch. This reusability inspires us to leverage the similarity of circuit embeddings generated by CircuitFusion to improve inference. **S4: Retrieval-augmented inference.** When applying CircuitFusion to downstream tasks of new circuits, we propose retrieving the most similar known circuits as references and calibrating the final predictions utilizing these retrieved circuits. This method not only improves the fine-tuning results but also enables zero-shot inference for circuits.

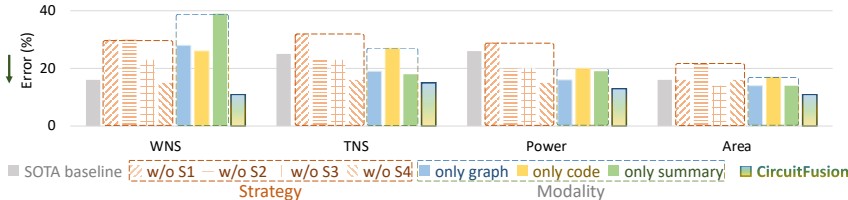

Figure 1: Preview of results on the effectiveness of proposed strategies and circuit modalities.

We evaluate the effectiveness of our proposed method across various tasks, where CircuitFusion's general solution consistently outperforms state-of-the-art (SOTA) task-specific models, demonstrating that CircuitFusion learns the truly general circuit embeddings applicable to various tasks. Additionally, as shown in Figure 1, we provide preview results on the impact of each proposed strategy and circuit modality, with more details provided in Appendix E.3. Specifically, relying on only a single circuit modality also leads to increased error. Similarly, removing any strategy results in a noticeable increase in prediction error, highlighting the effectiveness of our proposed multimodal fusion and circuit-specific strategies. Furthermore, we also evaluate the *scalability* of CircuitFusion by scaling both model size and data size, with details presented in Section 4.5. This suggests the potential for incorporating more diverse circuit datasets and larger models in future work. We believe our scalable and versatile CircuitFusion holds significant promise for enhancing other circuit-related tasks.

## 2 RELATED WORKS

**Hardware representation learning.** Several latest works have started exploring the learning of general circuit representations based on *pretrain-finetune* paradigms. However, these works (Wang et al., 2022; Li et al., 2022b; Xu et al., 2023; Shi et al., 2023) only focus on the circuit's *graph* format. Moreover, most studies (Wang et al., 2022; Li et al., 2022b; Shi et al., 2023) target the functional verification tasks for netlist, a late stage of the design flow. In comparison, we target learning circuit representation at the early RTL stage, when designers design circuit functionality with HDL code. This early stage offers more flexibility for design quality optimization. Additionally, there are also explorations for FPGAs (Sohrabizadeh et al., 2023) and analog circuits (Zhu et al., 2022).

**Multimodal representation learning.** Multimodal learning has achieved remarkable success in multiple other domains, such as vision-language learning (Radford et al., 2021; Li et al., 2021; 2022a; 2023a; Akbari et al., 2021; Bao et al., 2022) and graph-language learning (Yin et al., 2020; Gao et al., 2020). Among all multimodal learning applications, software code is the most similar to hardware circuits. Existing software encoders primarily target the summary and code modalities (Feng et al., 2020; Li et al., 2023b; Wang et al., 2023a; Zhang et al., 2024), while some (Guo et al., 2022a; Zhang et al., 2024) consider the syntactic graph format. However, multimodal software encoders cannot be directly applied to hardware tasks due to clearly different underlying mechanisms. For example, all four circuit properties (P1-P4) introduced in the Introduction do not apply to software code. Additionally, most existing software encoders are only validated on short code snippets within 1024 tokens, which are even shorter than the HDL of a simple circuit.

## 3 PROPOSED METHOD: CIRCUITFUSION

In this section, we present our CircuitFusion encoder framework in detail. We will first outline the circuit preprocessing steps and the model architecture, followed by an illustration of the pre-training process and its applications: 1) Circuit preprocessing (Section 3.1): We first split the entire circuit into multiple sub-circuits according to the parallel execution of sequential registers in circuits. Each sub-circuit is represented in three modalities: HDL code, circuit graph, and functionality summary. 2) Model architecture (Section 3.2): We encode the three circuit modalities via three unimodal encoders, with a multimodal encoder for modality fusion and an auxiliary netlist encoder to integrate implementation information. 3) Pre-training of CircuitFusion (Section 3.3 and Section 3.4): We propose 4 self-supervised pre-training tasks to jointly train CircuitFusion, achieving multimodal fusion and implementation-aware alignment to capture both structural and semantic information from circuits. 4) Application of CircuitFusion (Section 3.5): We propose a new retrieval-augmented inference method to improve CircuitFusion to downstream tasks.

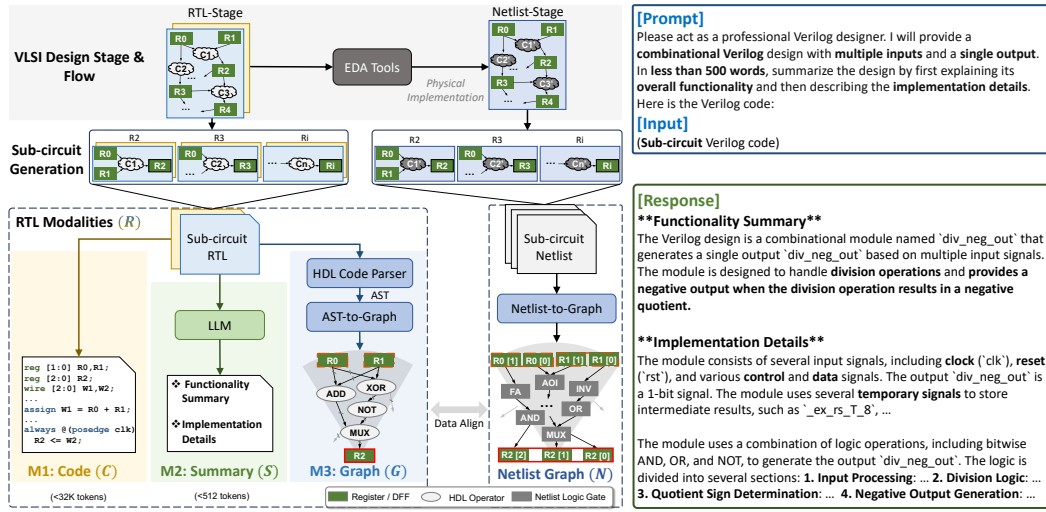

(a) Circuit preprocessing flow  (b) A prompt example on summary

Figure 2: Multimodal and multi-stage circuit preprocessing flow. We split circuits into sub-circuits for fine-grained encoding, representing RTL in three modalities (HDL code, functionality summary, and graph) and netlists as graphs, ensuring data alignment across modalities and design stages.

## 3.1 PREPROCESSING: MULTIMODAL AND MULTI-STAGE CIRCUIT DATA

Figure 2 illustrates our circuit data preprocessing workflow. For each circuit, we split it into multiple subcircuits, then express each subcircuit in three modalities (i.e., code, graph, summary). Each training circuit data is in both RTL and netlist, two primary circuit design stages. More preprocessing details with a concrete example of the circuit data can be found in Appendix B.

**Generation of three circuit modalities.** **(1) HDL code:** A circuit may be initially designed with different HDL code formats, such as Verilog, VHDL, and Chisel. For consistency, all HDL code is automatically converted into Verilog format with open-source tools (Wolf et al., 2013). **(2) Circuit graph:** To further capture circuit structure, each circuit in HDL code will be converted to a graph of logic operators and sequential registers. **(3) Functionality summary:** To further capture circuit semantics, as Figure 2b shows, we prompt GPT-4 to summarize the functionality and implementation details of each sub-circuit. We introduce the generation of sub-circuit below.

**Sub-circuit generation.** Utilizing the parallel execution mechanism of circuits, we split an entire circuit into multiple sub-circuits, each corresponding to one register. Specifically, for each register, we capture a sub-circuit by backtracing all its combinational *input* logic operators. This process applies to both RTL and netlist stages, and across all modalities, ensuring that the sub-circuits are consistently split and functionally aligned. The detailed sub-circuit generation algorithm is in Appendix B. CircuitFusion will encode each sub-circuit into an embedding, supporting various downstream tasks.

There are several key advantages of splitting circuits based on the register: (1) The sub-circuits for multimodal RTL and the cross-stage netlist are strictly aligned and functionally equivalent, ensuring consistency across both modalities and design stages. (2) Each sub-circuit captures the complete state transition of the register within a single clock cycle, including all timing paths and logic computations. This serves as a foundation for our model to learn both the combinational and sequential behavior of circuits. (3) The sub-circuit provides an intermediate level of granularity, bridging the gap between detailed tokens/nodes and the overall circuit. This granularity also enables generating summaries for the state update function of each register, offering a more fine-grained understanding compared to summarizing the entire circuit.

## 3.2 CIRCUITFUSION MODEL ARCHITECTURE

Figure 3 shows the model architecture of CircuitFusion. It consists of three unimodal encoders, one for each modality. Then a multimodal fusion encoder integrates the output information from all three unimodal encoders. Additionally, an auxiliary netlist encoder is employed only during pretraining.

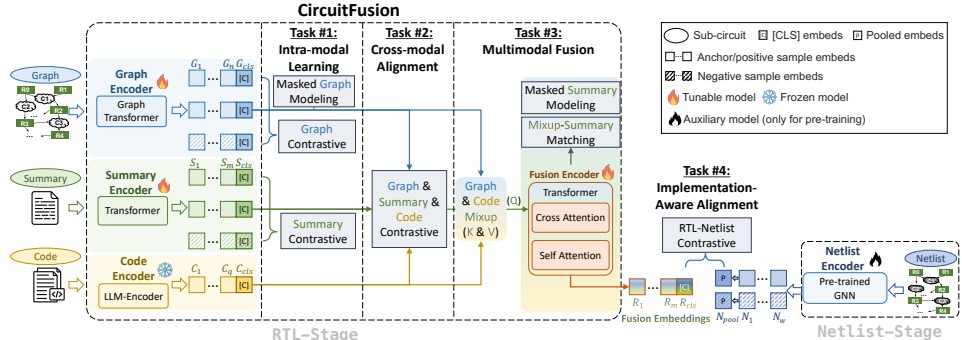

Figure 3: CircuitFusion pre-training workflow. CircuitFusion includes three unimodal encoders (graph, summary, and code) and a multimodal fusion encoder, with an auxiliary netlist encoder used only in pre-training. Leveraging the circuit's unique properties, we propose four tasks to capture structural and semantic information, while aligning the netlist stage for implementation awareness.

Here we introduce each unimodal encoder: **(1) Graph encoder:** Unlike text or image encoders (Devlin, 2018; Dosovitskiy, 2020) that benefit from extensive pre-trained models, pre-trained general-purpose graph models are relatively scarce, requiring us to build our graph encoder from scratch. We adopt a 7-layer graph transformer (Ying et al., 2021) with graph positional encoding. Each sub-circuit graph is encoded into a sequence of graph embeddings $\{G_{\text{cls}}, G_1, \ldots, G_n\}$, where $n$ is the number of nodes in the graph. **(2) Summary encoder:** We employ a 6-layer transformer to process the functionality summary with $m$ tokens into summary embeddings $\{S_{\text{cls}}, S_1, \ldots, S_m\}$, initialized using the first 6 layers of $\text{BERT}_{\text{base}}$ (Devlin, 2018). **(3) Code encoder:** To handle long HDL code snippets, we use an LLM-based text encoder NV-Embed-V1 (Lee et al., 2024), capable of a maximum of 32K input tokens. It encodes the HDL code with $q$ tokens into a sequence of code embeddings $\{C_{\text{cls}}, C_1, \ldots, C_q\}$. We freeze this code encoder to allow integration with various potential LLM-based encoders, including both open-source and commercial models, supporting scalability to larger models in the future. In all three encoders, the [CLS] token represents the embedding of the entire sub-circuit.

As for the **fusion encoder**, we initialize the multimodal encoder using the last 6 layers of $\text{BERT}_{\text{base}}$, equipped with the widely adopted cross attention mechanism for multimodal fusion (Li et al., 2021; 2022a; 2023a). We propose a summary-centric fusion to fuse the three modalities, since the summary provides higher-level semantic insights, while code and graph capture details of the circuit. More details of different fusion strategies are discussed in Appendix E.5. Specifically, we first perform a mixup (Zhang, 2017) of the graph node embeddings and code token embeddings, controlled by an interpolation coefficient $\lambda$. The resulting mixup embedding is defined as $\lambda\{G_1, G_2, \ldots, G_n\} + (1 - \lambda)\{C_1, C_2, \ldots, C_q\}$, where the two modality embedding vectors are padded into the same dimension first. The mixup strategy facilitates pre-fusion between the graph and code modality for better fusion with the summary modality. In the fusion encoder, the summary token embeddings are directly fed as queries for cross attention at each layer, while the mixup embeddings serve as the keys and values. Ultimately, the initial RTL sub-circuit is encoded into fused embeddings $\{R_{\text{cls}}, R_1, \ldots, R_m\}$.

In addition, an auxiliary GNN-based **netlist graph encoder** is already pre-trained on netlist before being applied to CircuitFusion. This auxiliary encoder encodes each sub-circuit netlist graph with $w$ nodes into node embeddings $\{N_1, N_2, \ldots, N_w\}$ and a graph-level embedding $N_{\text{pool}}$ by mean pooling all node embeddings. Please note that the netlist encoder is discarded during the inference process, as netlist data is unavailable in our RTL-stage downstream tasks. The implementation details of the netlist encoder are provided in Appendix C.3.

### 3.3 Pre-Training within CircuitFusion: Multimodal Fusion

Figure 3 also shows the pre-training process of CircuitFusion. To learn informative representations utilizing the unique circuit properties, we carefully design four pre-training tasks to train CircuitFusion: #1 Intra-modal learning on unimodal encoders to extract features within its specific modality. #2 Cross-modal contrastive learning for modality alignment. #3 Masked summary modeling and

mixup-summary matching on the fusion encoder for multimodal fusion. #4 Implementation-aware alignment between CircuitFusion and netlist encoder by cross-design-stage contrastive learning.

**Task #1 Intra-modal learning.** For the circuit graph modality, we first introduce the **masked graph modeling** pre-training objective. It captures the structural information of different circuit operators in relation to their connectivity within the circuit graph. Specifically, we randomly mask the graph nodes (i.e., operators) with the special token `[MASK]`. Then their operator type (e.g., ADD, OR, MUX, etc.) will be predicted based on surrounding graph node embeddings. Denote the input circuit masked graph as $\hat{G}$. The ground-truth operator type of masked nodes is denoted as a one-hot vector $\boldsymbol{y}_{\hat{G}}^{msk}$. The predicted operator type of masked nodes is denoted as $\boldsymbol{p}^{msk}(\hat{G})$, where the prediction is based on the surrounding graph node embeddings. The objective is to minimize the mean squared error (MSE) between $\boldsymbol{y}_{\hat{G}}^{msk}$ and $\boldsymbol{p}^{msk}$ as formulated below:

$$\mathcal{L}_{\text{MGM}}^{\#1} = \mathbb{E}_{(\hat{G}) \sim \mathcal{D}} \left[ \left( \boldsymbol{y}_{\hat{G}}^{msk} - \boldsymbol{p}^{msk}(\hat{G}) \right)^2 \right], \tag{1}$$

where $\mathbb{E}_{(\hat{G}) \sim \mathcal{D}}$ represents the expectation $\mathbb{E}$ over the circuit graph dataset $\mathcal{D}$.

Additionally, it's crucial to capture the functional semantics of each sub-circuit. To achieve this, we employ **intra-modal contrastive learning** for both the graph and summary encoders. Specifically, we augment each sub-circuit with positive samples (denoted as $^+$) generated through functionally equivalent transformations, which create new sub-circuits with the same functionality but entirely different structures. All other functionally different sub-circuits in the batch are treated as negative samples ($^-$). A contrastive objective pulls functionally similar circuits closer together in their respective modality embedding space while pushing dissimilar circuits further apart. Specifically, we minimize the InfoNCE (Oord et al., 2018) loss (defined as CL) for sub-circuit embeddings in both the graph ($G_{cls}$) and summary ($S_{cls}$) modalities, as formulated below:

$$\mathcal{L}_{\text{CL}_G}^{\#1} = \mathbb{E}_{(G) \sim \mathcal{D}}[\text{CL}(G_{\text{cls}}, G_{\text{cls}}^+, G_{\text{cls}}^-)], \quad \mathcal{L}_{\text{CL}_S}^{\#1} = \mathbb{E}_{(S) \sim \mathcal{D}}[\text{CL}(S_{\text{cls}}, S_{\text{cls}}^+, S_{\text{cls}}^-)]. \tag{2}$$

**Task #2 Cross-modal alignment.** We employ **cross-modal contrastive learning** to align graph, summary, and code representations. This task aligns representations from different modalities in a shared latent space, benefiting the subsequent modality fusion. We formulate the cross-modal contrastive loss as follows:

$$\mathcal{L}_{\text{CL}_{\text{modal}}}^{\#2} = \mathbb{E}_{(S,G,C) \sim \mathcal{D}} \left[ \text{CL}(S_{\text{cls}}, G_{\text{cls}}^+, G_{\text{cls}}^-) + \text{CL}(S_{\text{cls}}, C_{\text{cls}}^+, C_{\text{cls}}^-) \right]. \tag{3}$$

**Task #3 Multimodal fusion.** We adopt two pre-training tasks for modality fusion: (1) **Mask summary modeling:** This objective involves randomly masking parts of the high-level summary tokens with `[MASK]` and predicting them based on the fusion embeddings $R$. This helps the model capture the relationship between the modalities and reinforces its understanding of the summary. The model takes the masked summary embeddings $\hat{S}$ as the cross-attention query, and the graph-code mixup embeddings $mix_{GC}$ as the key and value. The probability for a masked token $\boldsymbol{p}^{msk}(\hat{R}$ is predicted by the model. This objective minimizes a cross-entropy (CE) loss:

$$\mathcal{L}_{\text{MSM}}^{\#3} = \mathbb{E}_{(\hat{S},G,C) \sim \mathcal{D}} \text{CE} \left( \boldsymbol{y}_{\hat{S}}^{msk}, \boldsymbol{p}^{msk}(\hat{R}) \right), \tag{4}$$

where $\boldsymbol{y}_{\hat{S}}^{msk}$ is ground-truth in one-hot vocabulary distribution.

(2) **Summary and mixup-embedding matching:** We employ a binary classification task that predicts whether a pair of given summary-mixup embeddings is positive (matched) or negative (unmatched). This task ensures that the model correctly fuses the summary with the mixup embeddings. Denote the binary prediction in probability as $\boldsymbol{p}^{match}$ and its ground-truth value as $\boldsymbol{y}^{match}$, we formulate the training objective as follows:

$$\mathcal{L}_{\text{match}}^{\#3} = \mathbb{E}_{(S,G,C) \sim \mathcal{D}} \text{CE} \left( \boldsymbol{y}^{match}, \boldsymbol{p}^{match}(R^+, R^-) \right). \tag{5}$$

### 3.4 Pre-Training beyond CircuitFusion: Implementation-Aware Alignment

**Task #4 Implementation-aware alignment.** Besides the above three tasks within CircuitFusion, we further pre-train CircuitFusion to integrate the later-stage netlist implementation information.

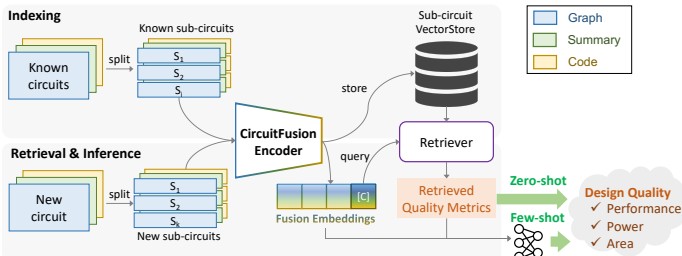

Figure 4: CircuitFusion retrieval-augmented inference flow. For downstream tasks, CircuitFusion retrieves the most similar known circuits as references to improve fine-tuning and enable zero-shot.

Specifically, we utilize an auxiliary netlist encoder with an implementation-aware contrastive objective, which aligns the embeddings of the whole sub-circuit in both the RTL (i.e., $R_{cls}$) and netlist (i.e., $N_{pool}$) within a shared latent space. The contrastive loss is formulated below:

$$\mathcal{L}_{CL_{impl}}^{\#4} = \mathbb{E}_{(R,N)\sim\mathcal{D}} \left[ \text{CL}(R_{cls}, N_{pool}^+, N_{pool}^-) + \text{CL}(N_{pool}, R_{cls}^+, R_{cls}^-) \right]. \tag{6}$$

Before the alignment, the netlist encoder is already pre-trained using masked graph modeling and graph contrastive objectives on netlist, with tasks similar to those employed in the CircuitFusion graph encoder. These objectives help the netlist encoder to capture the structure of the netlist graph.

To this end, we formulate the complete self-supervised pre-training objective of CircuitFusion by jointly employing the four tasks:

$$\mathcal{L} = (\mathcal{L}_{MGM}^{\#1} + \mathcal{L}_{CL_G}^{\#1} + \mathcal{L}_{CL_S}^{\#1}) + \mathcal{L}_{CL_{modal}}^{\#2} + (\mathcal{L}_{MSM}^{\#3} + \mathcal{L}_{match}^{\#3}) + \mathcal{L}_{CL_{impl}}^{\#4}. \tag{7}$$

### 3.5 APPLICATION: RETRIEVAL-AUGMENTED INFERENCE FOR DOWNSTREAM TASKS

When applying CircuitFusion to downstream tasks, we propose a retrieval-augmented inference method[2], leveraging the circuit reuse property, as illustrated in Figure 4. Given the similarity across different chip product generations and the extensive reuse of IP blocks, there is a vast pool of functionally similar known circuits that can be used as references. The key idea is to utilize the pre-trained CircuitFusion encoder to retrieve these similar circuits during inference on new designs, using their design quality metrics as references for downstream tasks.

Specifically, Figure 4 details the two steps: 1) **Indexing:** This step stores already known circuit[3] embeddings for future retrieval. The known multimodal circuits are first split into sub-circuits, converted into fusion embeddings by CircuitFusion, and stored in a VectorStore, denoted as $VS_{kn}$. 2) **Retrieval and inference:** For each new sub-circuit, the top-k most similar sub-circuits are retrieved from $VS_{kn}$ by measuring their cosine similarity in the embedding space.

During fine-tuning, the retrieved metrics (e.g., timing, power, area) from these similar circuits are concatenated with the fusion embeddings of the target unknown circuit and fed into fine-tuning models, which are trained with task-specific labels. These metrics act as additional reference points, allowing the fine-tuning model to calibrate and improve accuracy. Leveraging the retrieval-augmented strategy, the pre-trained CircuitFusion is fine-tuned with task labels by adding a simple regression model. Moreover, the retrieved metrics can directly serve as the final prediction, enabling zero-shot inference, where no further model training or task-specific labels are required. We provide more retrieval-augmented inference implementation details in Appendix C.5.

## 4 EXPERIMENTS

**Circuit dataset.** We construct a dataset with 41 RTL designs collected from various representative open-source benchmarks (Corno et al., 2000; URL, b; VexRiscv, 2022; Amid et al., 2020), with detailed statistics provided in Table 1. We provide more introductions on the benchmarks in Appendix A. The original dataset consists of 7,166 aligned RTL and netlist sub-circuit pairs, where each

---

[2]Please note this retrieval is optional during inference. If no similar design exists, omitting this strategy will result in only a slight performance decrease, with more detailed evaluations shown in Figure 9.

[3]In practice, known circuit designs may come from previous version of circuit product, IP libraries, etc.

Table 1: Statistics of Circuit Benchmarks.

| Source Benchmarks | # Circuit | Circuit Size {Min, Avg, Max} | | | Sub-circuit Size {Min, Avg, Max} | | Original HDL Type |
|---|---|---|---|---|---|---|---|
| | | # Node (Graph) | # Token (Code) | # Sub-circuit | # Node (Graph) | # Token (Code) | |
| ITC'99 | 7 | {3, 6, 9}K | {30, 70, 108}K | {12, 21, 31} | {12, 18, 1K} | {68, 1k, 2K} | VHDL |
| OpenCores | 5 | {1, 23, 54}K | {12K, 1M, 2M} | {12, 96, 173} | {8, 18, 1K} | {74, 2K, 91K} | Verilog |
| VexRiscv | 17 | {4, 147, 521}K | {144K, 6M, 20M} | {39, 168, 694} | {15, 47, 4K} | {68, 3K, 123K} | SpinalHDL |
| Chipyard | 12 | {1, 13, 83}K | {9K, 1M, 5M} | {28, 461, 23K} | {24, 50, 1K} | {70, 6K, 109K} | Chisel |

RTL sub-circuit is represented through three modalities. To enable contrastive learning, we perform circuit augmentation using functionally equivalent transformations through open-source tools like Yosys (Wolf et al., 2013) and ABC (Brayton & Mishchenko, 2010), generating positive samples and resulting in a total of 57,328 sub-circuits across the dataset. Theoretically, the dataset can be further augmented with an unlimited number of designs. We apply an 80/20 training/test split, ensuring the split is based on complete designs, with 33 designs used for training and 8 reserved for testing.

**Effectiveness of sub-circuit generation.** The effectiveness of our proposed sub-circuit generation is demonstrated in Table 1. The original circuit consists of tens of thousands of graph nodes and millions of code tokens, making it extremely challenging for existing graph and text models to handle. In comparison, sub-circuits are approximately 1000× smaller in the number of nodes and code tokens, thus enabling scalable and fine-grained representation learning.

## 4.1 VISUALIZATION OF CIRCUIT MULTIMODAL FUSION

Before delving into the quantitative experiment results, we first visualize the impact of cross-attention between the centric summary modality and corresponding code and graph modalities, as shown in Figure 5. Specifically, we use the Grad-CAM technique (Selvaraju et al., 2017) to highlight the graph nodes and code tokens with the highest cross-attention scores. For example, in Figure 5a, when the summary describes input signals, cross-attention highlights the corresponding signal names in both code and graph modalities. In Figure 5b, for conditional logic, both the relevant operations and signals are highlighted in the code modality, while conditional operations are highlighted in the graph modality. This visualization shows CircuitFusion's ability to focus on relevant elements across modalities: the graph provides structural information, the code offers semantic details, and the summary guides the fusion, enabling comprehensive representation learning.

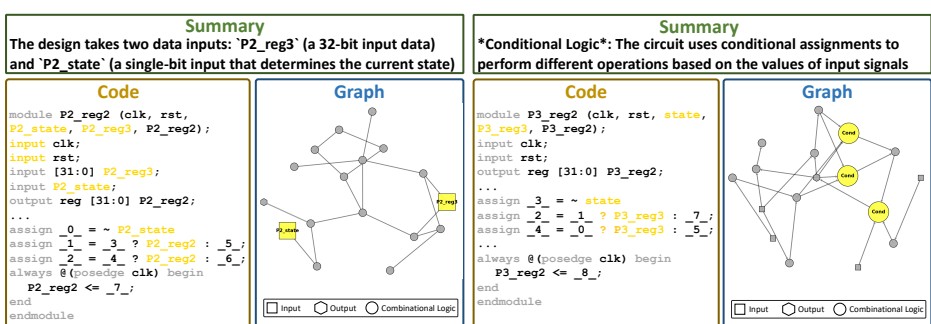

(a) Cross attention on input signals   (b) Cross attention on conditional logics

Figure 5: Visualization of cross attention between summary and code/graph.

## 4.2 DESIGN QUALITY PREDICTION TASKS AND BASELINE METHODS

CircuitFusion is pre-trained based on our proposed four self-supervised tasks, with implementation details in Appendix C. Then we apply pre-trained CircuitFusion to RTL-stage circuits, targeting five different prediction tasks about ultimate circuit qualities. These predictions are critical for designers to receive early feedback on design quality while writing HDL code, without going through time-consuming downstream design processes with EDA tools. These tasks are highly challenging, as the RTL-stage circuit only contains functional information, lacking the implementation details related to design quality. Recently, there have been several task-specific explorations (Sengupta et al., 2022; Xu et al., 2023; 2022; Fang et al., 2023; 2024b;a) on individual tasks.

**Design quality evaluation tasks at RTL stage.** We detail the five design quality prediction tasks as follows: (1) **Register slack prediction**: Predicting the slack for individual registers, which indicates

the margin by which the register meets or misses the timing constraints post-synthesis. This task helps identify critical registers that might cause timing violations. (2) **WNS prediction**: Estimating the worst negative slack (WNS) for the entire design, which measures the largest timing violation in the circuit. A key metric for determining whether the design meets timing requirements. (3) **TNS prediction**: Predicting the total negative slack (TNS), which sums up all the negative slack across the circuit. This metric indicates the overall severity of timing violations and helps prioritize timing optimization efforts. (4) **Total power prediction**: Estimating the total power consumption of the circuit, to assess the energy efficiency. (5) **Total area prediction**: Evaluating the total silicon area required to implement the circuit, which is critical for determining the physical feasibility and cost. Please note that the slack prediction task is directly evaluated at the sub-circuit level, whereas the other four tasks are ultimately evaluated on the entire circuit design. Moreover, we discuss the potential applications of CircuitFusion in multi-clock circuits in Appendix G, as well as optimization tasks and RTL generation in Appendix F, highlighting its adaptability and future research directions.

**Evaluation metrics.** We evaluate the accuracy with regression metrics including correlation coefficient ($R$) and mean absolute percentage error ($MAPE$) between labels and predictions.

**Baselines.** We comprehensively compare CircuitFusion against SOTA methods from diverse domains, including hardware task-specific solutions, general text encoders, and software code encoders. Hardware solution baselines include task-specific supervised methods `RTL-Timer` (Fang et al., 2024a) and `MasterRTL` (Fang et al., 2024b), as well as the self-supervised pre-trained circuit encoder `SNS v2` (Xu et al., 2023). Software solution baselines include recent pre-trained software code encoders, including `CodeSage` (Zhang et al., 2024), the encoder from `CodeT5+` (Wang et al., 2023a), and `UnixCoder` (Guo et al., 2022a), a multimodal code encoder. These models have an input token limit of 1024, which prevents them from directly processing the longer HDL code, and therefore we crop the input HDL code to fit this token limit. As for the general text encoders, we use `NV-Embed-V1` (Lee et al., 2024), with a 32k input token capacity, one of the top-performing text encoders on the MTEB LeaderBoard (Muennighoff et al., 2022).

### 4.3 SUPERVISED FINE-TUNING FOR DESIGN QUALITY TASKS

We first compare CircuitFusion against the aforementioned baselines across the five design quality prediction tasks. The detailed experimental results are demonstrated in Table 2.

Table 2: Comparison between CircuitFusion and baseline methods across all five design quality prediction tasks. CircuitFusion consistently outperforms all baseline methods, including hardware solutions, general LLM-based text encoders, and software code encoders across all tasks.

| Type | Method | Slack | | WNS | | TNS | | Power | | Area | |
|---|---|---|---|---|---|---|---|---|---|---|---|
| | | R | MAPE | R | MAPE | R | MAPE | R | MAPE | R | MAPE |
| Hardware Solution | RTL-Timer | 0.85 | 17% | 0.9 | 16% | 0.96 | 25% | N/A | | N/A | |
| | MasterRTL | N/A | | 0.89 | 18% | 0.94 | 28% | 0.89 | 26% | 0.98 | 16% |
| | SNS v2 | N/A | | 0.82 | 22% | N/A | | 0.76 | 28% | 0.93 | 25% |
| Text Encoder | NV-Embed-v1 | N/A | | 0.49 | 17% | 0.97 | 55% | 0.85 | 44% | 0.86 | 24% |
| Software Code Encoder | UnixCoder | N/A | | 0.46 | 21% | 0.95 | 44% | 0.83 | 29% | 0.85 | 26% |
| | CodeT5+ Encoder | N/A | | 0.55 | 21% | 0.63 | 43% | 0.49 | 46% | 0.45 | 39% |
| | CodeSage | N/A | | 0.23 | 25% | 0.86 | 45% | 0.8 | 38% | 0.77 | 41% |
| **Ours** | **CircuitFusion** | **0.87** | **12%** | **0.91** | **11%** | **0.99** | **15%** | **0.99** | **13%** | **0.99** | **11%** |

**Overall performance summary.** CircuitFusion consistently outperforms all the hardware solutions, general text encoders, and software code encoders across all five design quality prediction tasks. Its high correlation and low MAPE demonstrate its effectiveness and reliability in learning HDL code representations. Compared to SOTA baseline methods, CircuitFusion achieves a MAPE improvement of 5% for slack, WNS, and area prediction, 10% for TNS, and 13% for power prediction.

Unlike hardware solutions that often require intensive, task-specific modifications, CircuitFusion serves as a flexible foundation for multiple tasks, allowing fine-tuning without significant adjustments, which enhances its versatility. Moreover, the significant performance gap between CircuitFusion and text/software-based models highlights the importance of building hardware-specific models like CircuitFusion for tasks within the hardware design realm.

We conduct ablation studies to demonstrate the effectiveness of each modality and proposed strategy, as shown in Figure 1, with detailed evaluations provided in Appendix E.3. Furthermore, we apply our strategies to baseline methods and assess the improvements in Appendix E.4.

### 4.4 ZERO-SHOT RETRIEVAL AND REGRESSION

Compared with the existing hardware solutions, CircuitFusion is the first method to support zero-shot circuit quality prediction due to our innovative retrieval-augmented method. We provide a detailed evaluation of how the number of retrievals affects the prediction results, and we also compare it with the LLM-based and software code encoders. As shown in Table 3, CircuitFusion consistently performs best at top-1 retrieval for all tasks, achieving the lowest MAPE across the board. According to this result, we set the retrieval number to 1 in our retrieval-augmented inference to minimize error. Additionally, this result demonstrates that the encoded circuit embeddings contain rich semantic circuit information, enabling effective search and differentiation of functionally similar circuits. In contrast, other general text or software code encoders struggle to identify functionally similar HDL code snippets. This gap highlights the importance of integrating circuit-specific properties into hardware circuit learning. We also perform detailed few-shot fine-tuning experiments by increasing the training data from 0% (i.e., zero-shot) to 100%, as demonstrated in Appendix E.2.

Table 3: MAPE(%) results of the zero-shot top-k similar circuit retrieval.

| Method | Slack | | | | Sub-circuit Power | | | | Sub-circuit Area | | | |
|---|---|---|---|---|---|---|---|---|---|---|---|---|
| | top-1 | top-3 | top-5 | top-10 | top-1 | top-3 | top-5 | top-10 | top-1 | top-3 | top-5 | top-10 |
| LLM Encoder | 51 | 35 | 33 | 34 | 92 | 90 | 90 | 90 | 90 | 88 | 88 | 88 |
| UnixCoder | 56 | 36 | 36 | 36 | 90 | 89 | 90 | 91 | 89 | 88 | 89 | 89 |
| CodeT5+ Embedding | 57 | 35 | 35 | 36 | 88 | 87 | 89 | 90 | 87 | 86 | 87 | 88 |
| CodeSage | 50 | 36 | 36 | 36 | 89 | 87 | 88 | 91 | 88 | 85 | 86 | 87 |
| **Ours** | **21** | **22** | **23** | **26** | **36** | **40** | **42** | **53** | **35** | **40** | **42** | **51** |

### 4.5 DOWNSTREAM PERFORMANCE SCALING WITH MODEL SIZE AND DATA SIZE

In Figure 6, we study how the downstream task performance of CircuitFusion scales with both model size and pre-training data size. The plot shows the average performance across all five design quality prediction tasks after fine-tuning. The results indicate that increasing both model size and data size significantly enhances performance, demonstrating the scalability of CircuitFusion. This indicates the potential that further scaling of both model and data size in the future could lead to even greater improvements in accuracy and generalization.

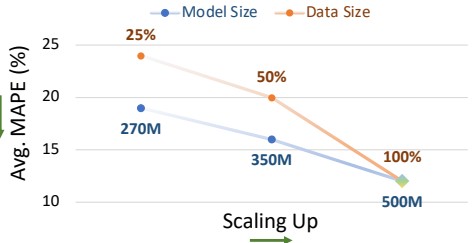

Figure 6: On the downstream task performance scaling with pre-trained CircuitFusion model size and data size.

Specifically, when the model size is increased from 270M to 500M parameters, the error decreases from 19% to 12%. This trend shows that larger models are able to capture more complex structural and semantic details from the circuit data, leading to better performance. Similarly, scaling the data size also leads to a noticeable reduction in error. Similarly, scaling the pre-training data size from 25% to 100% (i.e., 7k RTL sub-circuits) of the dataset reduces the error from around 24% to 12%, highlighting the importance of using a large, diverse circuit dataset for pre-training. We also provide model size statistics for the baseline methods as a comparison in Appendix E.1.

## 5 CONCLUSION AND FUTURE WORK

In this work, we propose CircuitFusion, the first multimodal fusion and implementation-aware circuit encoder tailored for hardware circuits. We introduce four innovative strategies that leverage the unique properties of circuits, spanning from circuit preprocessing to CircuitFusion's pre-training and application in downstream tasks. CircuitFusion is evaluated across five design quality prediction tasks, where it consistently achieves state-of-the-art performance after fine-tuning and even enables zero-shot inference.

**Limitations and future work.** While CircuitFusion is already trained in several different benchmarks and outperforms the existing hardware task-specific approaches, the amount of training data used is still limited compared to what would be ideal for a robust circuit foundation model. For future work, we aim to explore circuit data generation techniques and further scale CircuitFusion by pre-training on larger sets of synthetic circuit data. We will also release large-scale CircuitFusion as a foundation model to support different tasks for other users.

## ACKNOWLEDGEMENT

This work is supported by National Natural Science Foundation of China 62304192, Hong Kong Research Grants Council (RGC) CRF Grant C6003-24Y, and ACCESS – AI Chip Center for Emerging Smart Systems, sponsored by InnoHK funding, Hong Kong SAR.

## REPRODUCIBILITY STATEMENT

We provide references to relevant sections and materials to assist readers and researchers in replicating our results.

**Dataset description:** All datasets used in our experiments are from open-source benchmarks. A summary of these datasets is available in Appendix A, with a demonstration example shown in Appendix B.2. Detailed preprocessing methods are described in Appendix B, including the different circuit modality generation, sub-circuit generation, and downstream task label collection. The corresponding scripts can be found in our open-source repository.

**Open access to CircuitFusion code:** The source code for CircuitFusion is publicly available at: `https://github.com/hkust-zhiyao/CircuitFusion`. The repository includes scripts with step-by-step instructions to replicate the primary results presented in this paper.

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

## A    MORE ON CIRCUIT HDL DATASET

This section provides an overview of the various circuit HDL datasets used in our work, including ITC'99, OpenCores, VexRiscv, and Chipyard. These datasets offer diverse designs that span a range of hardware implementations, enabling comprehensive benchmarking of CircuitFusion across different circuit design tasks.

### A.0.1    ITC'99

The ITC'99 benchmark suite (Corno et al., 2000) is a widely used collection of hardware circuit designs, primarily designed for logic synthesis and verification. ITC'99 provides diverse designs ranging from simple combinational logic to more complex sequential circuits. VHDL

### A.0.2    OPENCORES

The OpenCores repository (URL, b) offers open-source hardware designs, including a wide variety of digital systems, such as CPUs, memory controllers, communication protocols, etc. OpenCores is a rich dataset for benchmarking HDL models because of its diverse collection of designs, which range from small, simple circuits to large, complex ones. Its open-source nature allows for flexibility in circuit modification, making it ideal for research and development in hardware design.

### A.0.3    VEXRISCV

VexRiscv (VexRiscv, 2022) is an open-source, RISC-V compliant CPU core designed using Spinal-HDL. This dataset focuses on CPU design and features a highly configurable architecture, allowing for variations in pipeline stages, instruction sets, and optimizations. The VexRiscv dataset is particularly useful for testing the scalability and flexibility of models in handling CPU-level design tasks, making it a valuable resource for benchmarking models like CircuitFusion on processor design tasks.

### A.0.4    CHIPYARD

Chipyard (Amid et al., 2020) is a comprehensive framework for building RISC-V-based system-on-chip (SoC) designs. It includes a collection of CPU cores, accelerators, memory systems, and I/O components, offering a complete design ecosystem for hardware developers. The Chipyard dataset enables testing at the SoC level, providing a broad set of circuits with varying complexities and design objectives.

## B    MORE ON CIRCUIT DATA PREPROCESSING

### B.1    DATASET AND FINE-TUNING LABEL COLLECTION

The labels for fine-tuning (i.e., post-synthesis PPA metrics) are generated through an industrial-standard logic synthesis process applied to benchmark RTL designs. In the open-source benchmarks, the HDL code of RTL circuits is provided, where the RTL stage describes the functional behaviors of the circuit. We then use the EDA tool Synopsys Design Compiler® to automatically synthesize the RTL circuits into gate-level netlists. The netlists represent real circuit implementations, consisting of logic gates (e.g., ADD, INV, AND, etc.) and registers (DFF). The synthesis process is performed with the open-source NanGate 45nm technology library (URL, a). Following the synthesis process in our baseline method (Fang et al., 2023), we use the "`compile_ultra`" command in Design Compiler to ensure that the synthesized netlists achieve high-quality PPA metrics at the Pareto frontier, with minimal variance across different configurations and optimization objectives. The design quality metrics of netlists are then obtained through Synopsys Prime Time® after synthesis, including slack of each register and WNS, TNS, total power, and total area for each RTL design.

### B.2    MULTIMODAL AND MULTI-STAGE CIRCUIT: A CASE STUDY

In this subsection, we provide a detailed example demonstrating the three modalities of RTL circuits.

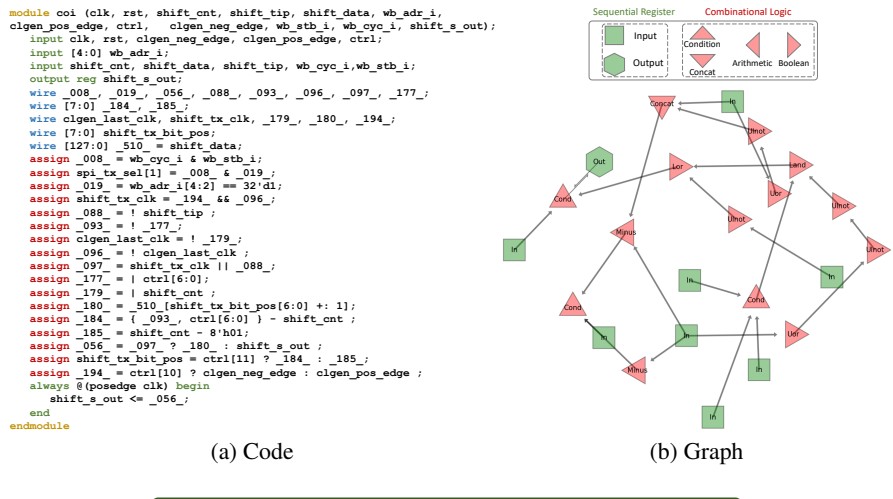

(a) Code           (b) Graph

**Functionality Summary**
The Verilog design, `shift_s_out`, is a combinational circuit that takes multiple inputs and produces a single output, `shift_s_out`. The circuit appears to be a part of a larger system, likely a **SPI (Serial Peripheral Interface) transmitter**, and is responsible for **shifting data out of a shift register**.
The circuit's functionality can be broken down into several key components:
1. **Shift Register Control**: The circuit takes input `shift_cnt` and uses it to control the shift register. It also takes input `shift_tip` to determine when to stop shifting.
2. **SPI Transmission**: The circuit generates a clock signal, `shift_tx_clk`, and uses it to shift data out of the shift register. It also generates a bit position signal, `shift_tx_bit_pos`, to select the current bit being transmitted.
3. **Control Signals**: The circuit takes input `ctrl` and uses it to control the shift register and SPI transmission. It also generates several control signals, such as `clgen_pos_edge` and `clgen_neg_edge`, which are used to control the clock signal.
4. **Data Shifting**: The circuit takes input `shift_data` and shifts it out of the shift register using the `shift_tx_clk` signal.

**Implementation Details**
The Verilog design uses a combination of assignment statements and an always block to implement the circuit's functionality.
* **Assignment statements** are used to define the relationships between the circuit's inputs and outputs. For example, `assign _008_ = wb_cyc_i & wb_stb_i;` defines the value of `_008_` as the logical AND of `wb_cyc_i` and `wb_stb_i`.
* The **always block** is used to update the `shift_s_out` output signal on the rising edge of the `clk` signal. The always block contains a single statement, `shift_s_out <= _056_;`, which assigns the value of `_056_` to the `shift_s_out` output signal.
* The design uses several **intermediate signals**, such as `_008_`, `_019_`, and `_180_`, to simplify the implementation and improve readability. These signals are used to compute the final output value, `shift_s_out`.

Overall, the design is well-structured and easy to follow, with clear and concise assignment statements and a simple always block.

(c) Summary

Figure 7: An example for multimodal circuit

### B.2.1 HDL CODE

As shown in Figure 7a, the HDL code for each sub-circuit is directly used as one of the input modalities, capturing the functional description of the circuit's behavior at the RTL stage. In this Verilog HDL code, a `module` represents an entire sub-circuit, where `input` and `output` specify the primary signals, `wire` connects the internal signals, `assign` represents combinational logic operations, and the `always` block triggered by a clock signal defines the behavior of sequential registers.

### B.2.2 STRUCTURAL GRAPH

Each sub-circuit HDL code is parsed into an abstract syntax tree (AST), which is then used to construct a control data flow graph, following a similar process in Fang et al. (2023). As demonstrated in Figure 7b, the nodes represent sequential registers and combinational operators (e.g., AND, ADD, EQUAL, MUX), while the wires connecting elements in the HDL code serve as the edges between these nodes.

### B.2.3 FUNCTIONALITY SUMMARY

We employ GPT-4o (Achiam et al., 2023) from Open-AI to summarize both the functionality and the implementation details of each sub-circuit HDL code. An example generated summary is illustrated in Figure 7c. A sub-circuit contains only combinational logic for a single register within a single clock cycle, making it simpler for the LLM to analyze without dealing with the complex sequential state transitions of the entire circuit.

### B.2.4 NETLIST GRAPH

We follow a similar widely adopted method (Wang et al., 2022) to convert netlists into the graph format. Specifically, register flip-flops (FF) and logic gates (e.g., AOI, INV, FA, AND) are treated as the nodes, and the wires connecting these gates form the edges of the graph.

### B.3 SUB-CIRCUIT GENERATION ALGORITHM

We convert the HDL code into sub-circuit code snippets and the circuit graph into corresponding sub-graphs, using the same sub-circuit generation method for both modalities to ensure functional alignment. The detailed splitting algorithm is provided in Algorithm 1. Specifically, for each register, we apply a breadth-first search starting from that register, backtracking through all connected combinational logic until reaching the related input/output registers. This process is highly parallelized within a design, ensuring minimal runtime.

---

**Algorithm 1** Sub-circuit generation($s$)

| | |
|---|---|
| 1: $V \leftarrow \{s\}$ | ▷ Set of visited nodes |
| 2: $Q \leftarrow \{s\}$ | ▷ Queue with start node |
| 3: $R \leftarrow \emptyset$ | ▷ Set to store registers and inputs |
| 4: **while** $Q \neq \emptyset$ **do** | |
| 5:     $u \leftarrow$ dequeue $Q$ | ▷ Current node |
| 6:     **for all** $v \in u$.outgoing **do** | |
| 7:         **if** $type(v) \in \{\text{reg}, \text{in}\}$ **then** | |
| 8:             $R \leftarrow R \cup \{v\}$ | ▷ Add register/input to set |
| 9:             **continue** | ▷ Skip to next node |
| 10:         **if** $v \notin V$ **then** | |
| 11:             $Q \leftarrow Q \cup \{v\}$ | ▷ Add unvisited node to queue |
| 12:             $V \leftarrow V \cup \{v\}$ | ▷ Mark node as visited |
| 13:             $v$.setParent($u$) | ▷ Set parent node |
| 14: **return** $R$ | ▷ Return set of all registers and inputs |

---

## C IMPLEMENTATION OF CIRCUITFUSION

### C.1 MODEL HYPERPARAMETERS

We first detail the hyperparameters for the proposed unimodal encoders: For the **Graph encoder**, we train a 7-layer graph transformer Graphormer (Ying et al., 2021) from scratch to capture the complex relationships in circuit graph semantics and structure. This encoder uses graph positional encodings supporting up to 256 in-degrees and out-degrees for centrality encoding, a maximum distance of 5 for spatial encoding, and an edge dimension of 12. It produces graph embeddings with a dimension of 768. The node features are represented by one-hot encoding of the node type, and the edge features are based on one-hot encoding of edge types, determined by the types of connected nodes. The encoder has a hidden dimension of 256 and utilizes 3 attention heads. For the **Code encoder**, we employ a frozen LLM-based general text encoder NV-Embed-V1 (Lee et al., 2024), which handles a maximum input size of 32K tokens. This model is based on Mistral-7B-v0.1 and was ranked No. 1 on the Massive Text Embedding Benchmark (MTEB) as of May 24, 2024. It generates embeddings with a dimension of 4096, which are then linearly projected to 768. As for the **Summary encoder**, it is initialized using the first 6 layers of BERT$_{\text{base}}$ (Devlin, 2018), following the setup in (Li et al., 2021).

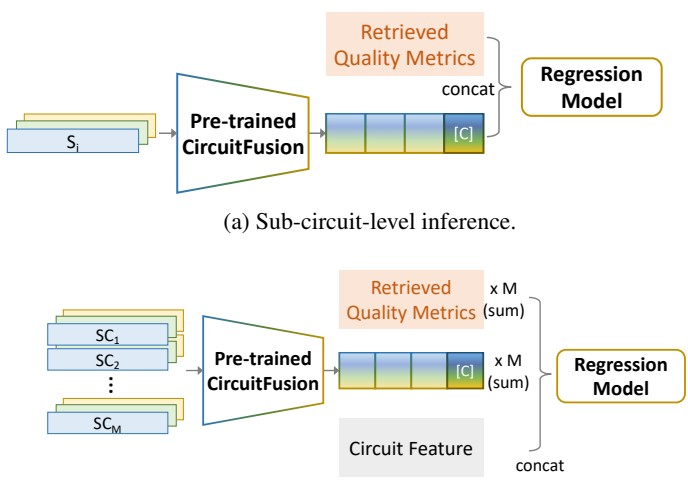

(a) Sub-circuit-level inference.

(b) Circuit-level inference.

Figure 8: Retrival-augmented inference implementation for tasks at different granularities.

For the **multimodal fusion encoder**, we initialize it with the last 6 layers of $\text{BERT}_{\text{base}}$. The fusion encoder is equipped with cross-attention mechanisms to enable the effective fusion of embeddings generated from the three unimodal encoders.

For the **auxiliary netlist encoder**, since graph transformers struggle with large and bit-blasted netlist graphs, we instead use a 3-layer GraphSAGE (Hamilton et al., 2017) GNN with a hidden dimension of 256. It encodes the netlist sub-circuit graphs into embeddings of 768 dimensions.

### C.2 SELF-SUPERVISED PRE-TRAINING TASKS IMPLEMENTATION

For masked graph modeling on both RTL and netlist sub-circuit graphs (used in Task #1 and netlist encoder), we adopt an approach inspired by GraphMAE (Hou et al., 2022). Specifically, 30% of the nodes in both the RTL and netlist graphs are randomly masked and reconstructed during each training epoch. A three-layer MLP with a hidden dimension of 256 is used to reconstruct the node types. Mean Squared Error (MSE) loss is applied to minimize the error between the original node type vectors and the reconstructed outputs, with the node types represented using one-hot encoding.

For the contrastive learning tasks (i.e., Task #1, #2, and #4), we utilize the InfoNCE loss function across all contrastive schemes. To balance the contributions of the different contrastive schemes, we assign an intra-modal weight of 1.0, while the cross-modal and implementation-aware weights are set to 0.2. The InfoNCE loss is formulated as follow:

$$NCE(E, E^+, E^-) = - \left[ \log \frac{\exp\left(\text{sim}(E, E^+)/\tau\right)}{\exp\left(\text{sim}(E, E^+)/\tau\right) + \sum_{E^-} \exp\left(\text{sim}(E, E^-)/\tau\right)} \right], \quad (8)$$

where $\tau$ is the temperature scaling parameter that controls the sharpness of the similarity scores, $E$ represents the circuit embeddings with positive samples ($E^+$) and negative samples ($E^-$). All temperature parameters are set to 0.3.

For the multimodal fusion tasks, including masked summary modeling and mixup embedding-summary matching, we adhere to the widely adopted tasks as described in (Li et al., 2021; 2022a; 2023a).

### C.3 NETLIST ENCODER IMPLEMENTATION

The auxiliary netlist encoder is pre-trained using the graph modality of gate-level netlists, as these netlists are flattened and contain limited semantics compared to RTL. The models and pre-training tasks are adapted from those in the graph encoder of CircuitFusion, with two key tasks: (1) Masked Graph Modeling: This task reconstructs the gate-level netlist nodes (e.g., logic gates) rather than RTL operators. (2) Graph Contrastive Learning: Positive samples are generated through Boolean equivalent transformations, ensuring structural variations retain functionality.

## C.4 TRAINING HYPERPARAMETERS

During the pre-training phase, the four self-supervised tasks are trained simultaneously for 50 epochs, with a total training time of approximately 20 hours. We use GELU as the activation function and set the batch size to 128. For optimization, we select AdamW, known for its ability to handle large-scale data effectively. The learning rate is warmed up to $1e-4$ during the first 1000 iterations, after which it follows a cosine decay schedule, gradually reducing to $1e-5$. This schedule ensures smooth convergence while avoiding abrupt gradient updates that could destabilize the training process.

In the fine-tuning phase, the pre-trained CircuitFusion model is frozen to preserve the learned representations, and lightweight models are applied to adapt to specific downstream tasks. To complement the learned sub-circuit representations, we integrate design-level features, such as the number of different operator types, to capture the overall design scale. Specifically, we explore various lightweight models, including additional MLP layers, GNN layers, and tree-based models like XGBoost. XGBoost consistently delivers the best performance due to its capability to efficiently handle the concatenation of sub-circuit embeddings with design-level features, treating them as tabular data.

Regarding the model development time, pre-training the model for 50 epochs takes approximately 10 hours using 4 Nvidia A4000 GPUs or around 32 hours on a single A4000 GPU. Fine-tuning is significantly faster, taking only around 5 minutes per task. Fine-tuning runtime efficiency is a key advantage of our method compared to task-specific solutions (i.e., our baselines). These task-specific methods often require time-consuming task-related feature engineering or model modifications, and lack generalizability to other tasks. In contrast, our pre-trained model allows developers to efficiently fine-tune for various design quality evaluation tasks, ensuring both flexibility and speed.

## C.5 RETRIEVAL-BASED INFERENCE IMPLEMENTATION

Our proposed retrieval-augmented inference process for CircuitFusion is illustrated in Figure 4. It is designed for two types of downstream tasks: sub-circuit-level and circuit-level.

**Sub-circuit-level inference:** For each sub-circuit, the pre-trained CircuitFusion encoder generates the corresponding multimodal embedding. We employ a retrieval process to fetch the most functionally similar sub-circuits from a vectorstore that contains previously seen circuits. These retrieved sub-circuits provide their design quality metrics, which are directly concatenated with the embedding generated by CircuitFusion. The concatenated feature vector is then fed into a regression model to predict the final design quality metric for the sub-circuit.

**Circuit-level inference:** At the circuit-level, the entire design is composed of multiple sub-circuits. Each sub-circuit is individually encoded by the CircuitFusion encoder, producing embeddings. Similar to the sub-circuit-level inference, we retrieve quality metrics for each sub-circuit from the vectorstore. The embeddings and retrieved quality metrics for all sub-circuits are added to generate a comprehensive circuit-level feature vector. We also concatenate this with design-level features (e.g., operator counts) to reflect the overall scale of the design. The combined feature vector is then fed into a regression model to predict circuit-level design quality metrics.

## C.6 BASELINE METHOD IMPLEMENTATION

For all the baselines, we directly employ the open-sourced code provided in their papers if available. For methods without open-source code, we carefully re-implement their approaches based on their published descriptions to ensure a fair comparison. Regarding the comparisons with baseline methods, we use the same pre-trained CircuitFusion model across all tasks, only fine-tuning different task-specific downstream models, following the standard pretrain-finetune paradigm.

# D EXPERIMENTAL SETTINGS

Our CircuitEncoder is implemented in Python, utilizing Pytorch and DGL (Wang, 2019) for self-supervised pre-training and model implementation. Experiments are conducted on a server equipped with a 2.9 GHz Intel Xeon(R) Platinum 8375C CPU and 512 GB RAM, with four NVIDIA A4000 GPUs for model pre-training.

# E  MORE EXPERIMENTAL RESULTS

## E.1  BASELINE MODELS (EXTENDED)

We summarize the baseline model size compared with CircuitFusion in Table 4.

Table 4: Pre-trained baseline model statistics.

| Model | Model Size | Embedding Dim. | Max Token | Training Data Source |
|---|---|---|---|---|
| NV-embed-V1 | 7B | 4096 | 32768 | Various text |
| UnixCoder | 125M | 768 | 1024 | Software code |
| Code T5+ Encoder | 110M | 768 | 1024 | Software code |
| CodeSage | 1.3B | 768 | 1024 | Software code |
| CircuitFusion | 500M (+7B frozen) | 768 | 32768 | Hardware circuit |

## E.2  ZERO-SHOT AND FEW-SHOT INFERENCE (EXTENDED)

Tables 5 to 9 illustrate the performance of CircuitFusion compared to SOTA baselines for zero-shot and few-shot learning on five design quality prediction tasks. The baseline method is selected as the top-performing model from all baselines in Table 2. The x-axis represents the fraction of training data used, ranging from zero-shot (0%) to full-shot (100%), while the y-axis shows the MAPE. These results demonstrate CircuitFusion's effectiveness in both zero-shot and few-shot settings, making it a versatile and reliable model for early-stage design quality prediction tasks, where access to large datasets is often restricted.

**Zero-shot.** Only CircuitFusion supports this zero-shot capability due to our innovative retrieval-augmented method. While the baselines do not provide predictions in the zero-shot setting, CircuitFusion achieves reasonable prediction accuracy without any training data, demonstrating its unique advantage.

**Few-shot.** CircuitFusion is particularly effective when training data is limited, which is crucial given the data availability challenges in hardware circuit design. As more training data is introduced (from 1/8 to full-shot), CircuitFusion consistently outperforms the baselines across all tasks, showing steeper performance improvements. It achieves lower MAPEs in nearly all cases, highlighting its superior ability to generalize and learn with minimal data.

Table 5: Few-shot results (MAPE) on slack prediction (Sub-Circuit-level).

| Task: Slack | 100% | 50% | 25% | 13% | 0% |
|---|---|---|---|---|---|
| SOTA (RTL-Timer) | 15% | 16% | 19% | 30% | N/A% |
| CircuitFusion | 12% | 14% | 16% | 19% | 21% |

Table 6: Few-shot results (MAPE) on WNS prediction (Circuit-level).

| Task: WNS | 100% | 50% | 25% | 13% | 0% |
|---|---|---|---|---|---|
| SOTA (RTL-Timer) | 16% | 29% | 36% | 43% | N/A |
| CircuitFusion | 11% | 17% | 18% | 25% | 27% |

Table 7: Few-shot results (MAPE) on TNS prediction (Circuit-level).

| Task: TNS | 100% | 50% | 25% | 13% | 0% |
|---|---|---|---|---|---|
| SOTA (RTL-Timer) | 25% | 35% | 49% | 74% | N/A |
| CircuitFusion | 15% | 24% | 41% | 52% | 59% |

Table 8: Few-shot results (MAPE) on Power prediction (Circuit-level).

| Task: Power | 100% | 50% | 25% | 13% | 0% |
|---|---|---|---|---|---|
| SOTA (`MasterRTL`) | 26% | 37% | 46% | 55% | N/A |
| CircuitFusion | 13% | 34% | 43% | 54% | 62% |

Table 9: Few-shot results (MAPE) on Area prediction (Circuit-level).

| Task: Area | 100% | 50% | 25% | 13% | 0% |
|---|---|---|---|---|---|
| SOTA (`MasterRTL`) | 16% | 33% | 46% | 56% | N/A |
| CircuitFusion | 11% | 30% | 45% | 51% | 58% |

### E.3 ABLATION STUDY

**Effectiveness of proposed strategies.** Figure 9 shows our ablation study by removing key components employed in CircuitFusion strategies. Removing the sub-circuit generation severely limits CircuitFusion's ability to handle large-scale circuits, leading to the most significant error increases across all tasks. Without this splitting, the model struggles to capture fine-grained circuit details, which is essential for tasks like slack prediction that require sub-circuit-level embeddings. We further assess the impact of each pre-training objective by selectively removing them. In every case, this leads to a clear rise in MAPE, indicating the importance of each pre-training task in enhancing both structural and semantic circuit understanding. Excluding retrieval-augmented inference results in a substantial increase in MAPE across all tasks. This highlights the significant role retrieval plays in enhancing fine-tuning performance by utilizing functionally similar existing circuits as references.

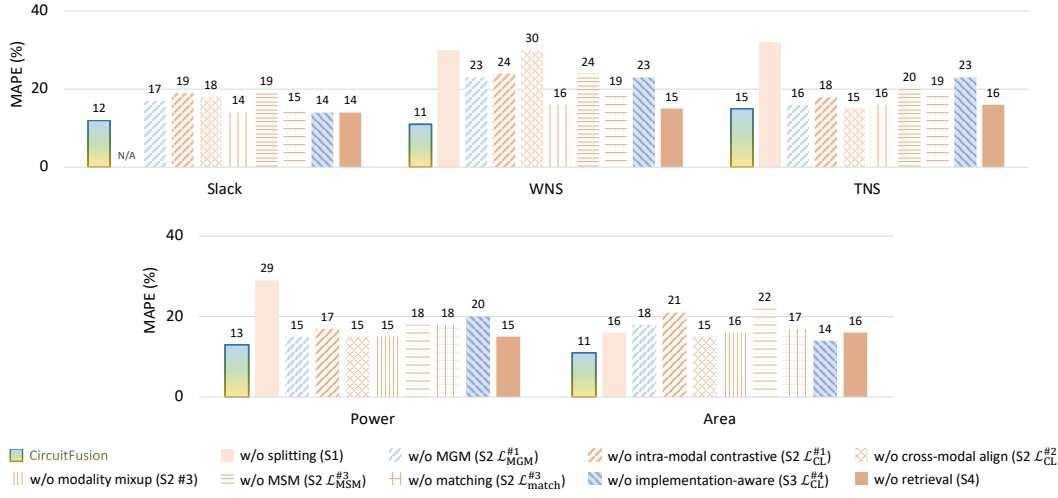

Figure 9: Ablation study on the effectiveness of proposed strategies.

**Impact of circuit modality.** In addition to the ablation study that evaluates the use of each modality individually in Figure 1, we also conduct an extended study on the selective removal of each modality. This study aims to further quantify the contribution of each modality (i.e., code, graph, and summary) to the model's overall performance. Specifically, when either the hardware code or graph modality is removed, there is a significant rise in prediction error across all tasks, highlighting their critical role in capturing both the structural and functional details of circuits. The graph modality, in particular, contributes more, as it contains rich structural information essential for circuit representation. These results demonstrate the necessity of leveraging modality fusion to fully capture the diverse characteristics of circuits.

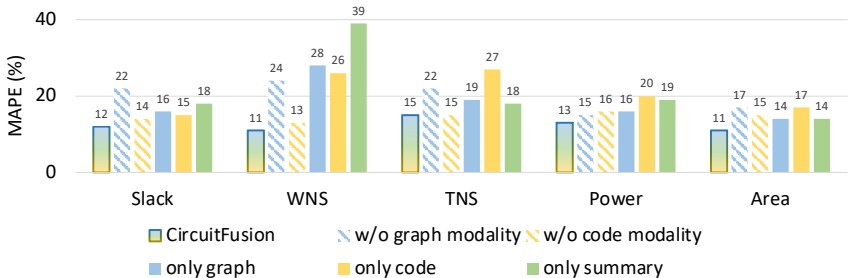

Figure 10: Ablation study on the impact of circuit modalities.

Table 10: Evaluation results when applying strategy S1 and S4 to other pre-trained encoders.

| Method | | Slack | | WNS | | TNS | | Power | | Area | |
|---|---|---|---|---|---|---|---|---|---|---|---|
| | | R | MAPE | R | MAPE | R | MAPE | R | MAPE | R | MAPE |
| NV-Embed-v1 | ori | N/A | | 0.49 | 26% | 0.97 | 55% | 0.85 | 44% | 0.86 | 24% |
| | w/ S1&4 | 0.85 | 15% | 0.81 | 17% | 0.95 | 27% | 0.99 | 20% | 0.97 | 17% |
| CodeSage | ori | N/A | | 0.23 | 21% | 0.86 | 45% | 0.8 | 38% | 0.77 | 41% |
| | w/ S1&4 | 0.84 | 14% | 0.9 | 25% | 0.95 | 24% | 0.96 | 18% | 0.96 | 17% |
| CodeT5+ Encoder | ori | N/A | | 0.55 | 30% | 0.63 | 43% | 0.49 | 46% | 0.45 | 39% |
| | w/ S1&4 | 0.83 | 14% | 0.8 | 21% | 0.94 | 24% | 0.95 | 19% | 0.93 | 21% |
| UnixCoder | ori | N/A | | 0.46 | 21% | 0.95 | 44% | 0.83 | 29% | 0.85 | 26% |
| | w/ S1&4 | 0.84 | 14% | 0.83 | 20% | 0.96 | 22% | 0.96 | 18% | 0.96 | 16% |
| **CircuitFusion** | | **0.87** | **12%** | **0.91** | **11%** | **0.99** | **15%** | **0.99** | **13%** | **0.99** | **11%** |

### E.4 APPLYING PROPOSED STRATEGIES TO BASELINE ENCODERS

As shown in Table 10, applying the sub-circuit generation (S1) and retrieval-augmented inference (S4) strategies to other pre-trained baseline encoders significantly boosts their performance across all tasks. By encoding sub-circuits instead of the entire circuit, all baseline methods are now able to handle the fine-grained slack prediction task, which they originally could not support.

For example, the LLM-based encoder NV-Embed-v1, despite its ability to process 32k tokens, struggles to encode entire circuit code sequences. When enhanced with S1 and S4, it achieves a notable reduction in MAPE for WNS (from 26% to 17%), TNS (from 55% to 27%), power (from 44% to 20%), and area (from 24% to 17%). Similarly, other software code encoders, such as CodeSage, CodeT5+ Encoder, and UnixCoder, also benefit significantly from these strategies. This shows that S1 and S4 not only improve fine-tuning accuracy but also enhance generalization across various design quality prediction tasks. Despite these improvements, CircuitFusion still outperforms all baselines, underscoring the effectiveness of its hardware-specific pre-training strategies.

### E.5 DISCUSSION ON DIFFERENT MULTIMODAL FUSION IMPLEMENTATIONS

In CircuitFusion, we propose a summary-centric fusion strategy, where graph and code embeddings, capturing detailed structural and semantic information, are first mixed. The Fusion Encoder then combines these mixup embeddings with summary embeddings to generate the final circuit representation. To evaluate the effectiveness of this strategy, we implemented multiple variants of CircuitFusion, including (1) other modality-centric (i.e., graph-centric and code-centric), (2) summary-centric partial fusion (i.e., summary+graph and summary+code), (3) simple alignment and inference with one modality (i.e., aligned graph, aligned code, and aligned summary), and (4) only single modality (i.e., graph, code, summary).

The experimental results, summarized in Table 11, demonstrate the superiority of CircuitFusion's summary-centric fusion strategy. It consistently outperforms other modality-centric, partial fusion, simple alignment, and single-modality approaches across all metrics, with significant improvements over SOTA baselines. These findings underscore the effectiveness of combining detailed structural and semantic information (from graph and code) with high-level functional insights (from summary) for comprehensive circuit evaluation.

Table 11: Evaluation results (MAPE%) for different multimodal fusion implementation variants.

| Variant Type | Method | Slack | WNS | TNS | Power | Area | *Avg.* |
|---|---|---|---|---|---|---|---|
| | **CircuitFusion** | **12** | **11** | **15** | **13** | **11** | **12** |
| 1. Other modality-centric | aligned graph-centric | 17 | 12 | 22 | 16 | 13 | 15 |
| | aligned code-centric | 20 | 16 | 16 | 16 | 11 | 16 |
| 2. Partial fusion | aligned summary+graph | 14 | 13 | 15 | 16 | 15 | 15 |
| | aligned summary+code | 22 | 24 | 22 | 15 | 17 | 20 |
| 3. Only modality alignment | aligned graph | 15 | 13 | 17 | 19 | 14 | 16 |
| | aligned code | 25 | 26 | 27 | 20 | 17 | 23 |
| | aligned summary | 16 | 21 | 15 | 14 | 15 | 16 |
| 4. Single modality | only graph | 16 | 28 | 19 | 16 | 14 | 19 |
| | only code | 25 | 26 | 27 | 20 | 17 | 23 |
| | only summary | 18 | 39 | 18 | 19 | 14 | 22 |
| | SOTA Baselines | 17 | 16 | 25 | 26 | 16 | 20 |

Table 12: Evaluation results on applying CircuitFusion for multi-clock circuit designs.

| Test Circuit | | | Slack | | WNS | TNS |
|---|---|---|---|---|---|---|
| Clock | Design | Frequency | R | MAPE | MAPE | MAPE |
| **Single-Clock** | itc1 | 1.5GHz | 0.91 | 6% | 5% | 8% |
| | chipyard1 | 1.5GHz | 0.88 | 12% | 16% | 15% |
| **Multi-Clock** | itc1 | 1.5GHz | 0.91 | 6% | 5% | 8% |
| | chipyard1 | 1GHz | 0.89 | 13% | 15% | 16% |

# F   APPLING CIRCUITFUSION FOR MORE AGILE CHIP DESIGN PROCESSES

In addition to the various design quality tasks supported by CircuitFusion, here we also discuss the application scenario of CircuitFusion for the agile chip design process. Inspired by (Fang et al., 2024a), these predictions can be further applied for early timing optimization, such as setting fine-grained timing optimization options for logic synthesis. Our better prediction results can seamlessly support the optimization method in [4] and would enable similar or even better optimization results.

Furthermore, a recent trend involves leveraging LLMs for direct RTL code generation and optimization (Liu et al., 2023b; Yao et al., 2024). Combining the multimodal RTL embeddings captured by our CircuitFusion encoder with such code generation decoders opens up a promising future research direction for designing and optimizing RTL code more efficiently.

# G   APPLING CIRCUITFUSION FOR MULTI-CLOCK CIRCUIT DESIGNS.

Currently, most AI-based timing evaluation methods (Wang et al., 2023b; Guo et al., 2022b; Fang et al., 2024a), including ours, primarily focus on circuits within a single clock domain (i.e., synchronous circuits). Multi-clock designs were not explicitly addressed in this work. However, our proposed S1 sub-circuit generation strategy inherently divides circuits into register cones by back-tracing logic to their driving registers. This process captures state transitions and all timing paths within each register cone. For multi-clock designs, timing predictions can be handled within individual clock groups by fine-tuning the model with timing labels specific to each group.

To demonstrate this capability, we combine two circuits from separate timing groups into a single design and perform logic synthesis. Then our CircuitFusion fine-tuned with different clock frequencies is used to predict the timing metric within each timing group. The prediction results with CircuitFusion are shown in Table 12. CircuitFusion achieves similar timing prediction accuracy to that observed in single-clock scenarios. However, challenges such as signal transfers across asynchronous clock domains (i.e., clock-domain crossing) are significantly more complex and currently fall beyond the scope of our method. This remains a promising direction for future exploration.

