# OpenReview forum: "CircuitFusion: Multimodal Circuit Representation Learning for Agile Chip Design"
_ICLR.cc/2025/Conference — ICLR 2025 Poster_

### Official Review · Reviewer_bMDX · 2024-10-30

**Soundness:** 3
**Presentation:** 3
**Contribution:** 3
**Rating:** 8
**Confidence:** 4

**Summary:**

This paper designs a CircuitFusion pre-training workflow to achieve an implementation-aware representation of RTL codes. The CircuitFusion aligns three RTL modalities(hardware code, structural graph, and functionality summary) with corresponding netlists and encodes them to an embedding vector for RTL design quality evaluation. The paper also presents a retrieval-augmented inference flow to improve fine-tuning and enable zero-shot.

I read the rebuttals and decided to keep my decision.

**Strengths:**

1. The CircuitFusion successfully encodes and aligns the hardware code, structural graph and functionality summary of RTL into an embedding vector and achieves SOTA performance. During pre-training, techniques include masked modeling, contrastive learning and mixup embedding summary matching are employed to enhance the model's understanding of circuit functionality.

2. This paper introduces a register-based circuit splitting method to enable fine-grained encoding and detailed representation with low computational complexity and scalability.

3. This paper proposes a retrival-augmented inference method to enable the model to integrate external knowledge effectively, leading to improvements in both zero-shot and few-shot learning scenarios.

4. Sufficient ablation studies and experimental evaluations are conducted to valitate the necessity of the above arrpoachs.

**Weaknesses:**

1. Regarding the technical novelty of this work, a related work was published on ICCAD 2024, named RTLRewriter [1]. Many techniques are similar to each other. RTLRewriter is a multimodal one that considers manual, code, and notes and also incorporates graph partitioning techniques, retrieval-augmented inference, etc.

2. During zero-shot inference, it is not reliable to simply average the performance metrics of retrieved similar circuits to make predictions. This issue seems to be reflected in the experimental results in Table 3, where the performance decreases with the increased available retrievals. Meanwhile, sufficient circuit data is required when building a VectorStore with quality metrics, which contradicts the concept of zero-shot inference.

3. The CircuitFusion implements summary-centric fusion to achieve alignment with information from other modalities. However, this approach may result in an underrepresentation of structural information or code-level details, which could be particularly important in timing, power and area performance. Additional experiments and explanations are needed to demonstrate the necessity of the summary mode.

4. The downstream applications should be further elaborated. Specifically, how to adjust the RTL design based on the obtained performance predictions warrants more detailed discussion. Currently, this work shows less relevance to the title's concept of 'agile chip design', as it does not adequately illustrate how CircuitFusion can be effectively applied to the optimization of RTL code. Prior works: [1] uses the encoded different modal RTL for new code reconstruction. [2] can notate detailed timing information on HDL and set fine-grained optimization options in the synthesis script for optimization applications.

5. The uploaded source code for CircuitFusion lacks readability and is hard to run. For example, many of the files mentioned in the README do not exist (data_bench, pretrain_model, pos and others).

[1] RTLRewriter: Methodologies for Large Models aided RTL Code Optimization [ICCAD 2024](https://arxiv.org/abs/2409.11414)

[2] Annotating Slack Directly on Your Verilog: Fine-Grained RTL Timing Evaluation for Early Optimization (https://arxiv.org/abs/2403.18453)

**Questions:**

1. This paper regards the performance of synthesized netlists as the RTL quality metrics. However, one RTL code can be synthesized into many gate-level netlists based on different configurations and optimization objectives, can the authors provide a detailed description on the selection of representative gate-level netlists, to represent the true performance of RTL and guide optimization.

2. In Table 3, please explain why the prediction error of CircuitFusion increases with the number of retrievals.

3. There is an inherent issue of information asymmetry between different circuit modalities, and it can be computationally expensive to calculate the similarity and alignment between these features. Can the authors provide more evidence to demonstrate that the summary-centric approach is more effective compared to simple alignment or using other modalities as the central focus?

---

> ### Author Response · Authors · 2024-11-19
> **Response to Reviewer bMDX (1/4)**
>
> We thank the reviewer for their insightful comments and for appreciating each proposed circuit-specific strategy in CircuitFusion and the extensive experiments. Below, we address your concerns, hoping our revisions clarify all the questions and strengthen the quality of our work.
>
>
> > Q1: Can the authors provide a detailed description on the selection of representative gate-level netlists, to represent the true performance of RTL and guide optimization.
>
> **A1**: Thanks for the question regarding the synthesis process. We also consider the impact of configurations and optimization objectives from the synthesis tool. We conducted an industrial-standard logic synthesis process applied to benchmark RTL designs. We have added detailed explanations about the dataset collection and labeling in **Appendix B.1**.
>
> Specifically, we use the widely-adopted open-source NanGate 45nm technology library and perform synthesis using the state-of-the-art industrial EDA tool, Synopsys Design Compiler. Following our baseline method [1], which analyzed synthesis options in Section IV-B, we utilize the industrial-standard "compile_ultra" command from the Design Compiler. This ensures that the PPA labels are optimized along the Pareto frontier and exhibit minimal variance across different configurations and objectives, as also demonstrated in Fig. 3 of [2]. After synthesis, we extract key design quality metrics, including register slack, WNS, TNS, total power, and total area, using Synopsys PrimeTime to ensure accurate and consistent labeling.
>
> [1] Wenji Fang et al. "MasterRTL: A Pre-Synthesis PPA Estimation Framework for Any RTL Design." In [ICCAD 2023](https://ieeexplore.ieee.org/document/10323951).
>
> [2] Jingyu Pan et al. "EDALearn: A Comprehensive RTL-to-Signoff EDA Benchmark for Democratized and Reproducible ML for EDA Research." In [arXiv](https://arxiv.org/abs/2312.01674).
>
> > Q2: In Table 3, please explain why the prediction error of CircuitFusion increases with the number of retrievals.
>
> **A2**: Thanks for the insightful question regarding the retrieval results. In Table 3, the prediction error increases with the number of retrievals because the most similar circuit retrieved by CircuitFusion typically has PPA metrics closest to the target circuit. As the number of retrievals grows, less similar circuits are included in the retrieved set, and the prediction error, calculated as the average of the retrieved results, increases. Based on this observation, we set the retrieval number to 1 in our retrieval-augmented inference to minimize error. We have updated the explanations in **Section 4.4**.

---

> ### Author Response · Authors · 2024-11-19
> **Response to Reviewer bMDX (2/4)**
>
> > Q3: Can the authors provide more evidence to demonstrate that the summary-centric approach is more effective compared to simple alignment or using other modalities as the central focus?
>
> **A3**: Thanks for the insightful question regarding the fusion strategy of CircuitFusion. As requested by the reviewer, we implement multiple variants of CircuitFusion, including
> - (1) Other modality-centric (i.e., graph-centric and code-centric)
> - (2) Summary-centric with one other modality (i.e., summary+graph and summary+code)
> - (3) Simple alignment and inference with one modality (i.e., aligned graph, aligned code, and aligned summary)
> - (4) Only single modality (i.e., graph, code, summary)
>
> We have also updated this result in Appendix E.5. The experimental results (MAPE%) of these CircuitFusion variants are demonstrated in the table below:
>
> | Variant   Type |      Method      | Slack | WNS | TNS | Power | Area | *Avg.* |
> |:--------------:|:----------------:|:-----:|:---:|:---:|:-----:|:----:|:----:|
> |                |   **CircuitFusion**  |   **12**  |  **11** |  **15** |   **13**  |  **11**  |  **12**  |
> |        1       |   Aligned Graph-centric  |   16  |  12 |  18 |   16  |  13  |  15  |
> |        1       |   Aligned Code-centric   |   20  |  16 |  16 |   16  |  11  |  16  |
> |        2       |   Aligned Summary+graph  |   14  |  13 |  15 |   16  |  15  |  15  |
> |        2       |   Aligned Summary+code   |   22  |  24 |  22 |   15  |  17  |  20  |
> |        3       |  Aligned graph   |   15  |  13 |  17 |   19  |  14  |  16  |
> |        3       |   Aligned code   |   25  |  26 |  27 |   20  |  17  |  23  |
> |        3       | Aligned summary  |   16  |  21 |  15 |   14  |  15  |  16  |
> |        4       |    Only graph    |   16  |  28 |  19 |   16  |  14  |  19  |
> |        4       |    Only code     |   25  |  26 |  27 |   20  |  17  |  23  |
> |        4       |   Only summary   |   18  |  39 |  18 |   19  |  14  |  22  |
> |                |  SOTA Baselines  |   17  |  16 |  25 |   26  |  16  |  20  |
>
> The results highlight the superiority of CircuitFusion’s summary-centric fusion strategy. Compared to other modality-centric, partial fusion, simple alignment, and single-modality approaches, CircuitFusion achieves the best performance across all metrics, with significant improvements over SOTA baselines. Specifically for the different modality-centric variants, the graph-centric and code-centric fusion strategies are slightly less effective, but still outperform the SOTA baseline methods. Additionally, the modality alignment also significantly improves the performance of each single modality (i.e., variant 3 vs. variant 4). All these experiments demonstrate the effectiveness of our key idea of multimodal circuit learning. Regarding the detailed modality fusion strategy, it can be further explored and refined in future works.
>
> In addition to the experimental support, our proposed summary-centric fusion strategy is intuitively motivated by the nature of the three modalities: the graph and code modalities provide detailed information (e.g., operator types, signal names), and the summary modality offers high-level functional semantics for the circuit. By first mixing the detailed modalities and then fusing them with the high-level summary modality, the reconstruction tasks (e.g., masked summary modeling) focus on leveraging the detailed information to reconstruct the semantics in the summary.

---

> ### Author Response · Authors · 2024-11-19
> **Response to Reviewer bMDX (3/4)**
>
> > W1: Regarding the technical novelty of this work, a related work was published on ICCAD 2024, named RTLRewriter [3]. Many techniques are similar to each other.
>
> **R1**: Thank you for pointing out this related work. While there are some overlapping elements to handle RTL circuits, our work and RTLRewriter are fundamentally different. We summarize the different contribution aspects below:
> - **Motivation and Application**: RTLRewriter targets RTL code generation and rewrite task, whereas our work aims to encode RTL code into embeddings for evaluating circuit design quality (i.e., PPA) across various tasks. They are two significantly distinct application focuses.
> - **AI model**: RTLRewriter directly employ the GPT4 model from OpenAI and employs prompts and RAG techniques to achieve the task, without developing their own AI model. In contrast, we develop the architecture of CircuitFusion from scrath, and design self-supervised objectives to pre-train it, and apply it for downstream tasks.
> - **Graph partitioning**: RTLRewriter partitions circuits based on predicted synthesis runtime for effective optimization, while CircuitFusion divides sequential circuits into register cones based on register boundaries to capture timing paths and state transitions.
> - **Retrieval**: RTLRewriter retrieves *textual documents* for RTL code *generation task* following the RAG technique. Our approach retrieves *similar circuits* to improve inference performance for *evaluation tasks*, focusing on enhancing predictions rather than generation. Our retrieval method is innovative and has no similar explorations before.
> - **Circuit modality**: RTLRewriter uses modalities like manual notes, code text, and diagram images. Diagram images are limited to tiny circuit demos and are unsuitable for our benchmarks, which involve thousands of operators. In contrast, we handle circuits as operator graphs, code text, and functionality summaries to ensure scalability and usability for large sequential circuits.
>
> Furthermore, RTLRewriter is a very recent contribution, submitted to arXiv on **September 4, 2024**, which is just three weeks before our submission deadline (**October 1, 2024**). And the ICCAD 2024 (**October 27, 2024**) conference was held after our ICLR submission deadline.
>
> These distinctions highlight the novelty and unique focus of our work compared to RTLRewriter. We have also added this related work in our introduction of AI for circuit applications in **Section 1**. The shared interest also emphasizes the growing importance of exploring multimodal fusion for RTL circuits and advancing AI techniques in the circuit design domain.
>
> [3] Xufeng Yao et al. "RTLRewriter: Methodologies for Large Models aided RTL Code Optimization." In [ICCAD 2024](https://arxiv.org/abs/2409.11414).
>
>
> > W2: During zero-shot inference, it is not reliable to simply average the performance metrics of retrieved similar circuits to make predictions. Meanwhile, sufficient circuit data is required when building a VectorStore with quality metrics, which contradicts the concept of zero-shot inference.
>
> **R2**: Thanks for the suggestion. We agree with the reviewer that relying solely on zero-shot inference (i.e., averaging the performance metrics of retrieved circuits) is not reliable for final predictions. Please note that the zero-shot inference is highly challenging, since there is even no task type information. But we are the first to investigate the potential of zero-shot PPA predictions.
>
> In our approach, the zero-shot retrieval process serves as an initial step for PPA prediction tasks. The retrieved circuit PPA metrics are then utilized as value features during fine-tuning, further enhancing the final prediction performance, as demonstrated in our ablation study (Figure 1, w/o S4). Table 3 primarily highlights that CircuitFusion embeddings effectively differentiate similar and dissimilar circuits, enabling the use of the most similar retrieved circuit to support fine-tuning.
>
> Regarding the circuit VectorStore, it is pre-built using existing circuit designs, such as prior versions of circuit products, IP libraries, and our pre-training dataset. This emphasizes the circuit reusability property (P4) summarized in Section 1. We describe the retrieval process as zero-shot inference because it operates without requiring label collection or fine-tuning during retrieval, aligning with the intended purpose of zero-shot methodologies.

---

> ### Author Response · Authors · 2024-11-19
> **Response to Reviewer bMDX (4/4)**
>
> > W3: Additional experiments and explanations are needed to demonstrate the necessity of the summary mode.
>
> **R3**: Please refer to **A3** above for detailed experiments and explanations.
>
> > W4: The downstream applications should be further elaborated.
>
> **R4**: Thanks for the valuable suggestions, we have added more detailed elaborations on the potential application scenarios in **Appendix F**. Regarding the examples [3,4] mentioned by the reviewer, currently, we evaluate CircuitFusion on the register slack prediction task proposed by [4], where CircuitFusion outperforms RTL-Timer [4], as shown in Table 2. For further applications based on the prediction, such as setting fine-grained timing optimization options for logic synthesis, our better prediction results can seamlessly integrate into the optimization framework in [4], enabling comparable or even superior optimization results.
>
> Regarding RTL code generation tasks as explored in [3], our work primarily focuses on encoding RTL circuits for design quality evaluation tasks, while [3] centers on the decoder component for code generation. RTL code generation is highly challenging and beyond the current scope of our encoder. However, combining the multimodal embeddings captured by CircuitFusion with code generation frameworks is a promising future research direction.
>
> [4] Wenji Fang et al. "Annotating Slack Directly on Your Verilog: Fine-Grained RTL Timing Evaluation for Early Optimization." In [DAC 2024](https://dl.acm.org/doi/10.1145/3649329.3655671).
>
>
>
> > W5: The uploaded source code for CircuitFusion lacks readability and is hard to run. For example, many of the files mentioned in the README do not exist (data_bench, pretrain_model, pos and others).
>
> **R5**: Thanks for pointing this out. The missing files mentioned (e.g., data_bench, pretrain_model, pos) correspond to circuit datasets and pre-trained models, which are too large to include in the GitHub repository. As the circuits we used are sourced from open benchmarks, these files can be generated by following the preprocessing scripts provided in the repository. We have updated the [README](https://anonymous.4open.science/r/CircuitFusion-EB45/README.md) file in the anonymous repository to clarify this.

---

> ### Author Response · Authors · 2024-11-27
> **Official Comment by Authors**
>
> Dear Reviewer bMDX,
>
> Thank you again for your valuable feedback and for taking the time to evaluate our work. We greatly appreciate your thoughtful insights and suggestions, which have helped us refine our manuscript and clarify our contributions.
>
> We have provided detailed responses to all the questions and concerns raised during the review process and have made corresponding updates to the paper to address these points. As today is the final day for paper revisions, we would greatly appreciate it if you could let us know whether our responses have adequately addressed your concerns or if there are any additional points we can clarify.
>
> Thank you once again for your time and consideration.
>
> Best regards,
>
> Authors

---

> ### Author Response · Authors · 2024-12-01
> **Official Comment by Authors**
>
> Dear Reviewer bMDX,
>
> We sincerely appreciate your thoughtful insights and suggestions on our work. With the extended rebuttal deadline approaching in 3 days, we would be most grateful if you could kindly share any additional feedback or highlight areas that might require further clarification or improvement. We remain committed to addressing your comments promptly and thoroughly.
>
> Thank you once again for your time and effort in reviewing our submission. Your guidance and consideration are greatly appreciated.
>
> Best regards,
>
> Authors

---

### Official Review · Reviewer_KB11 · 2024-11-03

**Soundness:** 3
**Presentation:** 3
**Contribution:** 3
**Rating:** 6
**Confidence:** 3

**Summary:**

This paper proposed a multimodal and implementation-aware circuit encoder. It encodes circuits into general representations that support different downstream circuit design tasks with three modalities: hardware code, structural graph, and functionality summary. CircuitFusion demonstrates excellent generalizability and the ability to learn circuits’ inherent properties compared to state-of-the-art supervised methods in the empirical study.

**Strengths:**

• This paper proposes the first multimodal fused and implementation-aware circuit encoder that fuses three modalities (i.e., HDL code, structural graph, and functionality summary).
• CircuitFusion outperforms state-of-the-art (SOTA) task-specific supervised models in design quality evaluation task empirically
• CircuitFusion is able to perform few-shot and zero-show inference.

**Weaknesses:**

• Most existing multi-modality encoders, such as CLIP, seem to be mainly used as a controller for generation, such as stable diffusion text to image generation. This paper, however, uses it as an evaluator, which I personally feel does not fit the purpose.
• I feel the argument about scalability is not significant enough to consider as one of the strengths. Since deep learning models are naturally scaled with larger datasets, and such large datasets are not available right now and may not be available in the future due to the IP issue, achieving excellent results with limited data is more important than scalability to large but nonexistent datasets.

**Questions:**

Could you provide examples from related work where multimodal models have been successfully used for evaluation tasks, particularly in hardware or software domains? Additionally, could you elaborate on the specific advantages that a multimodal approach offers for circuit evaluation tasks compared to traditional unimodal methods?

---

> ### Author Response · Authors · 2024-11-19
> **Response to Reviewer KB11 (1/2)**
>
> We thank the reviewer for their insightful comments and for appreciating the novelty of CircuitFusion and other strengths of our work. We address your concerns below, hoping our revisions clarify all the questions and strengthen the quality of our work.
>
> > W1 & Q1: This paper uses the multi-modality encoder as an evaluator, which I personally feel does not fit the purpose of existing encoders (e.g., CLIP). Could you provide examples from related work where multimodal models have been successfully used for evaluation tasks, particularly in hardware or software domains?
>
> **A1**: Thanks for this insightful question. We understand the reviewer's concern that existing representative multimodal encoders, particularly those for text and image modalities (e.g., CLIP, ALBEF), are primarily designed to align embeddings in a shared latent space. Then they are applied to *cross-modality tasks* such as text-to-image retrieval or generation control.
>
> In contrast, our work fuses three circuit modalities to generate final circuit embeddings for evaluation tasks, leveraging the multimodal inherent of circuits. As for the cross-modality purpose mentioned by the reviewer, we adopt this key idea to achieve the implementation-aware alignment, which aligns the embeddings from the RTL encoder (i.e., our CircuitFusion) and netlist encoder within a shared latent space to enable the netlist implmentation awareness at the RTL stage.
>
> Using AI models as evaluators is a well-established approach in circuit design, especially for circuit quality evaluation. These evaluators provide ultra-fast early stage feedback for circuit designers, supporting the agile chip design process. Recent explorations in the hardware domain have demonstrated the effectiveness of multimodal models in evaluating design metrics during physical design (e.g., placement and routing) stages. For example:
> - **Multimodal circuit learning for layout timing evaluation**: [1] fuse the netlist graph and layout image by concatenating the embeddings from a GNN and CNN as the final circuit embeddings, and use them for post-routing layout timing prediction. Ablation studies are also conducted to demonstrate the improvement brought by multimodal fusion.
> - **Multimodal circuit learning for layout congestion evaluation**: Similarly, [2] grafts netlist-based (i.e., GNN for netlist graph) message passing on a layout-based model (i.e., Swin Transformer for layout image) to improve congestion prediction performance. The authors also conducted ablation studies to demonstrate the enhancement through multimodal fusion.
>
> These two works combine two modalities to generate final circuit embeddings, similar to the purpose of our approach. The key difference lies in the circuit stages targeted. Specifically, our work focuses on RTL circuits, emphasizing logic functionality. To achieve both structural and semantic understanding, we fuse the operator graph, code text, and functionality summary. In contrast, their works target circuit layout stages, focusing on physical characteristics by fusing the netlist graph with layout images for structural and positional insights.
>
> In addition to evaluation tasks, a recent work (submitted to arXiv on 4 Sep 2024) mentioned by Reviewer bMDX combines multimodal circuit information (e.g., code text and diagram images) to enhance circuit semantic understanding, benefiting RTL code generation.
>
> In the software domain, [4,5] propose fusing software code text with data flow graphs or abstract syntax trees to enhance code representation learning for downstream tasks like Clone Detection and Code Search, which are similar to evaluation tasks in the hardware domain.
>
>
> [1] Ziyi Wang et al. "Restructure-Tolerant Timing Prediction via Multimodal Fusion." In [DAC 2023](https://ieeexplore.ieee.org/document/10247802).
>
> [2] Su Zheng et al. "Lay-Net: Grafting Netlist Knowledge on Layout-Based Congestion Prediction." In  [ICCAD 2023](https://ieeexplore.ieee.org/abstract/document/10323800).
>
> [3] Xufeng Yao et al. "RTLRewriter: Methodologies for Large Models aided RTL Code Optimization." In  [ICCAD 2024](https://arxiv.org/abs/2409.11414).
>
> [4] Daya Guo et al. "GraphCodeBERT: Pre-training Code Representations with Data Flow." In [ICLR 2021](https://arxiv.org/abs/2009.08366).
>
> [5] Daya Guo et al. "UniXcoder: Unified Cross-Modal Pre-training for Code Representation." In [ACL 2022](https://arxiv.org/abs/2203.03850).

---

> ### Author Response · Authors · 2024-11-19
> **Response to Reviewer KB11 (2/2)**
>
> > Q2: Could you elaborate on the specific advantages that a multimodal approach offers for circuit evaluation tasks compared to traditional unimodal methods?
>
> **A2**: Thanks for the question regarding the advantages of multimodal fusion. The multimodal approach provides a mult-view understanding of circuits. In evaluation tasks like timing analysis, while the graph structure is crucial, the functional semantics of different operators in timing paths also significantly influence results.
>
> For RTL circuits, the graph and code modalities capture detailed structural and semantic information, while the summary modality offers a high-level functional overview. Fusing these modalities enhances CircuitFusion’s understanding of RTL circuits. Similarly, for circuit layout as in [1,2], the graph modality provides structural details, and the image modality captures positional information, improving task performance.
>
>
> > W2: I feel the argument about scalability is not significant enough to consider as one of the strengths. Since deep learning models are naturally scaled with larger datasets, and such large datasets are not available right now and may not be available in the future due to the IP issue, achieving excellent results with limited data is more important than scalability to large but nonexistent datasets.
>
> **R**: Thanks for the suggestion regarding our scalability analysis. We agree with the reviewer that deep learning models are naturally scaled with larger datasets. Our scalability experiments were conducted to *quantitatively* evaluate the model’s ability to scale during pre-training. In addition to data size scaling, we also demonstrate model size scaling, where increasing model parameters significantly improves task performance.
>
> We also agree with the reviewer that circuit data availability remains a critical challenge due to privacy concerns. To address this challenge, our proposed strategies, such as multimodal fusion of diverse circuit data modalities and retrieval-augmented inference, are specifically designed to maximize the utility of limited datasets. These approaches enable CircuitFusion to achieve robust performance with constrained data resources, ensuring practical applicability.

---

> ### Comment · Reviewer_KB11 · 2024-11-25
>
> Thanks a lot for your response. Since I am not an expert in this field and I am not too confident to judge some of the intuition behind the multimodality technique used in this paper, I decided to keep my score as 6.

---

### Official Review · Reviewer_kA13 · 2024-11-03

**Soundness:** 2
**Presentation:** 3
**Contribution:** 3
**Rating:** 6
**Confidence:** 1

**Summary:**

This paper introduces an approach to circuit representation learning using multimodal fusion and implementation-aware alignment.

**Strengths:**

* The paper is structured well and easy to follow. All training steps and the proposed methodology is clearly illustrated with representative figures, these make it easy to follow and understand the proposed approach and results.
* Evaluations are comprehensive and comparison are done against many recent methods.
* The approach of this fusion approach that overcomes the hardware unique (P1 to P4) looks novel and interesting.

**Weaknesses:**

* The dataset contains only 41 RTL designs, this might not be representative of the entire space. It would be nice to demonstrate the effectiveness of the proposed method on a wider range of dataset.

**Questions:**

* For sequential logic, how well does it handle logics with multiple clocks? Can it optimize such tasks?

---

> ### Author Response · Authors · 2024-11-19
> **Response to Reviewer kA13 (1/2)**
>
> We thank the reviewer for their insightful comments and for appreciating the novelty of our hardware unique strategies, clear illustration, and comprehensive evaluations. In the responses below, we have carefully addressed each of the reviewer's questions.
>
> > Q1: For sequential logic, how well does it handle logics with multiple clocks? Can it optimize such tasks?
>
> **A1**: Thanks for the insightful question regarding the application of CircuitFusion to multi-clock designs. Currently, most AI-based timing evaluation methods [1-9], including ours, primarily focus on circuits within a single clock domain (i.e., synchronous circuits). We did not explicitly address multi-clock designs in this work.
>
> Although not specifically designed for multi-clock circuits, our proposed S1 sub-circuit generation strategy inherently divides circuits into register cones by backtracing logic to their driving registers. This captures state transitions and all timing paths within each register cone. For multi-clock designs, timing predictions can be handled within individual clock groups by fine-tuning with timing labels specific to those groups.
>
> To conduct the experiment on this scenario, since we do not include multi-clock designs in our benchmark, we combine two circuits from separate timing groups into a single design and perform logic synthesis. Then our CircuitFusion fine-tuned with different clock frequencies is used to predict the timing metric within each timing group. The results are shown in the table below,  CircuitFusion achieves similar timing prediction accuracy to that observed in single-clock scenarios. This experiment has also been detailed in **Appendix G**.
>
> |      Test   Circuit      |           |           | Slack |      |  WNS |  TNS |
> |:------------------------:|:---------:|:---------:|:-----:|:----:|:----:|:----:|
> |           Clock          |   Design  | Frequency |   R   | MAPE | MAPE | MAPE |
> | Single-Clock       |    itc1   |   1.5GHz  |  0.91 |  6%  |  5%  |  8%  |
> |                          | chipyard1 |   1.5GHz  |  0.88 |  12% |  16% |  15% |
> |        Multi-Clock       |    itc1   |   1.5GHz  |  0.91 |  6%  |  5%  |  8%  |
> |                          | chipyard1 |    1GHz   |  0.89 |  13% |  15% |  16% |
>
> However, challenges like signal transfers across asynchronous clock domains (i.e., clock-domain crossing) are significantly more complex and currently beyond the scope of our method. We hope this addresses your question.
>
> [1] Erick Carvajal Barboza et al. "Machine Learning-Based Pre-Routing Timing Prediction with Reduced Pessimism." In [DAC 2019](https://ieeexplore.ieee.org/document/8807063).
>
> [2] Zizheng Guo et al. "A timing engine inspired graph neural network model for pre-routing slack prediction." In [DAC 2022](https://dl.acm.org/doi/10.1145/3489517.3530597).
>
> [3] Xu He et al. "Accurate timing prediction at placement stage with look-ahead RC network." In [DAC 2022](https://dl.acm.org/doi/10.1145/3489517.3530598).
>
> [4] Ceyu Xu et al. "SNS's not a synthesizer: a deep-learning-based synthesis predictor." In [ISCA 2022](https://dl.acm.org/doi/10.1145/3470496.3527444).
>
> [5] Prianka Sengupta el al. "How Good Is Your Verilog RTL Code? A Quick Answer from Machine Learning." In [ICCAD 2022](https://ieeexplore.ieee.org/document/10069028).
>
> [6] Ziyi Wang et al. "Restructure-Tolerant Timing Prediction via Multimodal Fusion." In [DAC 2023](https://ieeexplore.ieee.org/document/10247802).
>
> [7] Wenji Fang et al. "MasterRTL: A Pre-Synthesis PPA Estimation Framework for Any RTL Design." In [ICCAD 2023](https://ieeexplore.ieee.org/document/10323951).
>
> [8] Cuyu Xu et al. "Fast, Robust and Transferable Prediction for Hardware Logic Synthesis." In [MICRO 2023](https://dl.acm.org/doi/10.1145/3613424.3623794).
>
> [9] Wenji Fang et al. "Annotating Slack Directly on Your Verilog: Fine-Grained RTL Timing Evaluation for Early Optimization." In [DAC 2024](https://dl.acm.org/doi/10.1145/3649329.3655671).

---

> ### Author Response · Authors · 2024-11-19
> **Response to Reviewer kA13 (2/2)**
>
> > W1: The dataset contains only 41 RTL designs, this might not be representative of the entire space. It would be nice to demonstrate the effectiveness of the proposed method on a wider range of datasets.
>
> **R1**: Thanks for the critical suggestion regarding the dataset. As discussed in Section 5 (Limitations) of the manuscript, access to high-quality open-source RTL designs is limited due to the proprietary nature of industrial circuits.
>
> Despite this, our dataset of 41 RTL designs is sourced from widely recognized open-source benchmarks, detailed in Table 1 and Appendix A. These designs cover four mainstream HDL types and diverse application domains, including logic synthesis, CPU cores, memory controllers, and communication protocols. These representative benchmarks are consistent with the hardware baseline methods we compared against. Additionally, by applying our S1 sub-circuit generation strategy, we expand the dataset into 57k sub-circuits, enabling comprehensive training of CircuitFusion.
>
> Regarding future work, current RTL code generation techniques using LLMs primarily produce simple combinational modules and cannot generate realistic, large-scale sequential circuits suitable for our methods. We plan to explore advanced RTL generation techniques, such as graph generation models, to create more complex and representative datasets.

---

> ### Author Response · Authors · 2024-11-27
> **Official Comment by Authors**
>
> Dear Reviewer kA13,
>
> Thank you again for your valuable feedback and for taking the time to evaluate our work. We greatly appreciate your thoughtful insights and suggestions, which have helped us refine our manuscript and clarify our contributions.
>
> We have provided detailed responses to all the questions and concerns raised during the review process and have made corresponding updates to the paper to address these points. As today is the final day for paper revisions, we would greatly appreciate it if you could let us know whether our responses have adequately addressed your concerns or if there are any additional points we can clarify.
>
> Thank you once again for your time and consideration.
>
> Best regards,
>
> Authors

---

> > ### Author Response · Authors · 2024-12-01
> > **Official Comment by Authors**
> >
> > Dear Reviewer kA13,
> >
> > We sincerely appreciate your thoughtful insights and suggestions on our work. With the extended rebuttal deadline approaching in 3 days, we would be most grateful if you could kindly share any additional feedback or highlight areas that might require further clarification or improvement. We remain committed to addressing your comments promptly and thoroughly.
> >
> > Thank you once again for your time and effort in reviewing our submission. Your guidance and consideration are greatly appreciated.
> >
> > Best regards,
> >
> > Authors

---

### Official Review · Reviewer_uzeD · 2024-11-04

**Soundness:** 3
**Presentation:** 3
**Contribution:** 3
**Rating:** 6
**Confidence:** 5

**Summary:**

This paper introduces a multi-modal representation method for agile digital IC design.
Evaluations show that the proposed method achieves significant performance improvement over several SOTA methods.

**Strengths:**

The paper proposes to use multimodal representations to estimate digital IC performance efficiently. This strategy is novel as compared to previous LLM-based methods.

The technique details on how to encode each modality and fuse modalities are elaborated clearly.

The paper comprehensively evaluates multiple tasks and compares them with many SOTA methods to show the impressive performance of the proposed method.

An ablation study is also performed to show the impact of each individual modality on the estimation error.

**Weaknesses:**

It is unclear why a summary-centric fusion is used to fuse the three modalities, lacking quantitative support.

It is unclear how the netlist encoder is pre-trained.

The comparisons with previous methods are unfair since the reported results are not based on the same number of model parameters across all baselines.

**Questions:**

Q1: How do you label data for fine-tuning?

Q2: How long does it take for pre-training and fine-tuning the model?

Q3: How are the correlation coefficient (R) and mean absolute percentage error (MAPE)in Table 2 obtained? I.e, Based on a single inference or multiple inferences?

---

> ### Author Response · Authors · 2024-11-19
> **Response to Reviewer uzeD (1/3)**
>
> We thank the reviewer for their insightful comments and for appreciating the novelty of our multimodal circuit fusion, clear writing, and extensive experiments. Below, we address your concerns, hoping our revisions clarify all the questions and strengthen the quality of our work.
>
> > Q1: How do you label data for fine-tuning?
>
> **A1**: We acknowledge the reviewer's question regarding the detailed circuit label generation process. The labels for fine-tuning (i.e., post-synthesis PPA metrics) are generated through an industrial-standard logic synthesis process applied to benchmark RTL designs. We have added detailed explanations about the dataset collection and labeling in **Appendix B.1**.
>
> Specifically, we use the widely-adopted open-source NanGate 45nm technology library and perform synthesis using the state-of-the-art industrial EDA tool, Synopsys Design Compiler. To ensure that the PPA labels are optimal at the Pareto frontier, we use the industrial-standard "compile_ultra" command in the Design Compiler, which minimizes PPA variance across different configurations and optimization objectives. After synthesis, we extract key design quality metrics, including register slack, WNS, TNS, total power, and total area, using Synopsys PrimeTime to ensure accurate and consistent labeling.
>
> > Q2: How long does it take for pre-training and fine-tuning the model?
>
> **A2**: Thanks for the question regarding model development time. We acknowledge that efficiency is crucial for users during fine-tuning. Pre-training the model for 50 epochs takes approximately 10 hours using 4 Nvidia A4000 GPUs or around 32 hours on a single A4000 GPU. Fine-tuning is significantly faster, taking only around 5 minutes per task. We have added the illustration in **Appendix C.4**.
>
>
>
>
>
> > Q3: How are the correlation R and MAPE in Table 2 obtained? I.e, Based on a single inference or multiple inferences?
>
> **A3**: Thanks for the question regarding evaluation metrics. These metrics strictly follow the hardware baseline methods. The correlation coefficient R and MAPE in Table 2 are calculated based on a single inference, as the results are fixed during inference. The calculations depend on the task type:
> - **Register-level slack prediction task:** Since each design contains numerous registers, we calculate R and MAPE for each design individually and report the average across all test designs in Table 2.
> - **Design-level tasks** (WNS, TNS, power, and area prediction): Each design has a single label. We calculate the R and MAPE metrics across all test designs.
>
> Given the label $Y_i$ and the prediction $\hat{Y} _ i$, the definition of R and MAPE is demonstrated as follows:
> $R = \frac{\sum_{i=1}^{n} (\hat{Y} _ i - \bar{\hat{Y}})(Y_i - \bar{Y})}{\sqrt{\sum_{i=1}^{n} (\hat{Y} _ i - \bar{\hat{Y}})^2 \sum_{i=1}^{n} (Y_i - \bar{Y})^2}}, \text{MAPE} = \frac{1}{n} \sum_{i=1}^{n} \left| \frac{Y_i - \hat{Y}_i}{Y_i} \right| \times 100\%.$

---

> ### Author Response · Authors · 2024-11-19
> **Response to Reviewer uzeD (2/3)**
>
> > W1: It is unclear why a summary-centric fusion is used to fuse the three modalities, lacking quantitative support.
>
> **R1**: Thanks for this insightful suggestion on implementing fusion strategies. To provide quantitative support, we conduct more experiments with variants of CircuitFusion, where the other two modalities are used as the centric focus, and we have added the experiments and discussion in **Appendix E.5**:
> - **Graph-Centric Fusion**: For model architecture, we replaced the current fusion transformer (i.e., BERT) with the graph transformer from the graph encoder, while keeping other components unchanged. The fusion graph transformer employs cross-attention, where graph embeddings are used as queries (Q), and the mixed embeddings from code and summary modalities serve as keys (K) and values (V). The two pre-training tasks for the Fusion Encoder are modified accordingly to: (1) Masked Graph Modeling, where masked graph operators are reconstructed, and (2) Mixup-Graph Matching, which matches mixed embeddings to their corresponding graphs.
> - **Code-Centric Fusion**: Since the code modality is also represented in text, the model architecture remained the same as in CircuitFusion. The only difference was making the code modality the centric focus, such that the other two modalities fused into the code embeddings. The two pre-training tasks for the Fusion Encoder are modified accordingly to: (1) Masked Code Modeling, where masked code tokens are reconstructed, and (2) Mixup-Graph Matching, which matches mixed embeddings to their corresponding code.
>
> The experimental results (MAPE%) for these variants are summarized below:
>
> |     Method     | Slack | WNS | TNS | Power | Area | *Avg.* |
> |:--------------:|:-----:|:---:|:---:|:-----:|:----:| :----:|
> |  **CircuitFusion** |   **12**  |  **11** |  **15** |   **13**  |  **11**  |  **12**  |
> |  Graph-centric |   16  |  12 |  18 |   16  |  13  |  15  |
> |  Code-centric  |   20  |  16 |  16 |   16  |  11  |  16  |
> | SOTA Baselines |   17  |  16 |  25 |   26  |  16  |  20  |
>
> Experimental results show that the summary-centric fusion (i.e., CircuitFusion) achieves the best performance across all tasks. The graph-centric and code-centric fusion strategies are slightly less effective, but still outperform the baseline methods. These experiments demonstrate the effectiveness of our key idea of multimodal circuit learning. Regarding the detailed modality fusion strategy, it can be further explored and refined in future works.
>
> In addition to the experimental support, our proposed summary-centric fusion strategy is intuitively motivated by the nature of the three modalities: the graph and code modalities provide detailed information (e.g., operator types, signal names), and the summary modality offers high-level functional semantics for the circuit. By first mixing the detailed modalities and then fusing them with the high-level summary modality, the reconstruction tasks (e.g., masked summary modeling) focus on leveraging the detailed information to reconstruct the high-level semantics in the summary.

---

> ### Author Response · Authors · 2024-11-19
> **Response to Reviewer uzeD (3/3)**
>
> > W2: It is unclear how the netlist encoder is pre-trained.
>
> **R2**: Thank you for pointing out this unclear explanation. We apologize for any confusion caused in our original writing. We have added the explanation of netlist encoder pre-training in **Appendix C.3**.
>
> To clarify, the netlist encoder is pre-trained using the graph modality of gate-level netlists, as these netlists are flattened and contain limited text semantics compared to RTL. The models and pre-training tasks are adapted from those in the graph encoder of CircuitFusion, with two key tasks:
> - **Masked Graph Modeling**: This task reconstructs the gate-level netlist nodes (e.g., logic gates) rather than RTL operators.
> - **Graph Contrastive Learning**: Positive samples are generated through Boolean equivalent transformations, ensuring structural variations retain functionality.
>
>
> > W3: The comparisons with previous methods are unfair since the reported results are not based on the same number of model parameters across all baselines.
>
> **R3**: Thanks for the suggestion. We acknowledge the reviewer's concern regarding the implementation of the baseline methods. Regarding the comparisons with baseline methods, we use the same pre-trained CircuitFusion model across all tasks, only fine-tuning different task-specific downstream models. This approach follows the standard pretrain-finetune paradigm. We have added the illustration in **Appendix C.6**.
>
> For the baselines, we directly employ the open-sourced code provided in their papers if available. For methods without open-source code, we carefully re-implement their approaches based on their published descriptions to ensure a fair comparison. As shown in Table 4 in the paper, these baselines inherently differ in the number of model parameters due to their unique architectures. While we acknowledge this difference number of parameters, it reflects the original designs of the baseline methods and aligns with the respective authors’ intent. We hope this clarification addresses your concerns.

---

> ### Comment · Reviewer_uzeD · 2024-11-23
> **Thanks for the response!**
>
> Thanks for the response. I have a follow-up question. Could you please explain why this multi-modal LLM does not show significant improvement than SOTA, given its multi-modality?

---

> > ### Author Response · Authors · 2024-11-23
> > **Response to Reviewer uzeD (follow-up question)**
> >
> > We kindly thank the reviewer for the discussion. Below, we address the concern regarding the improvement achieved by our multimodal CircuitFusion. Please let us know in case of any additional questions.
> >
> > > Q: Could you please explain why this multi-modal LLM does not show significant improvement than SOTA, given its multi-modality?
> >
> > **Quantitative Improvement.** In the experimental results, our CircuitFusion demonstrates notable improvement over SOTA baselines, achieving an average MAPE reduction of 8% across all metrics, as shown in the table (MAPE%) below:
> >
> > |     Method     | Slack | WNS | TNS | Power | Area | *Avg.* |
> > |:--------------:|:-----:|:---:|:---:|:-----:|:----:| :----:|
> > |  **CircuitFusion** |   **12**  |  **11** |  **15** |   **13**  |  **11**  |  **12**  |
> > | SOTA Baselines |   17  |  16 |  25 |   26  |  16  |  20  |
> >
> > Specifically, we observe notable improvements of **5%** in **slack** prediction, **5%** in **WNS**, **10%** in **TNS**, **5%** in **power**, and **7%** in **area**.
> >
> >
> > **New Pre-trained Paradigm Outperforms Task-Specific Baselines.** Predicting PPA metrics at the RTL stage is inherently challenging since these metrics rely on post-synthesis physical characteristics, such as gate delay, slew, and toggle rate—none of which are available at the RTL stage. The improvements achieved by CircuitFusion are particularly noteworthy given the task’s difficulty and the limitations of SOTA methods, which require task-specific modifications such as unique preprocessing, feature engineering, specialized model architectures, and re-training for each PPA prediction task.
> >
> > In contrast, CircuitFusion requires only lightweight fine-tuning on pre-trained embeddings. While SOTA methods already achieve high accuracy, our pre-trained approach consistently outperforms each task-specific SOTA method, delivering ultra-fast and more precise early-stage PPA feedback for RTL designs.
> >
> >
> > **Effectiveness of Multimodal Circuit Learning.** Existing SOTA baselines address this gap by leveraging only the graph modality of RTL designs. In comparison, CircuitFusion integrates RTL code, graph structure, and high-level summaries, effectively fusing all available RTL-stage information. Our multimodal approach builds a robust pre-trained model capable of providing more accurate early predictions. The effectiveness of multimodal alignment and fusion strategies are also demonstrated in the ablation experiments (MAPE%) required by Reviewer bMDX, as shown below. Specifically, integrating simple alignment (i.e., variant 3) and alignment with fusion (i.e., variant 1&2) both clearly improve the prediction accuracy compared with a single modality (i.e., variant 4) across all PPA metrics.
> >
> >
> > | Variant   Type |      Method      | Slack | WNS | TNS | Power | Area | *Avg.* |
> > |:--------------:|:----------------:|:-----:|:---:|:---:|:-----:|:----:|:----:|
> > |                |   **CircuitFusion**  |   **12**  |  **11** |  **15** |   **13**  |  **11**  |  **12**  |
> > |        1       |   Aligned Graph-centric  |   16  |  12 |  18 |   16  |  13  |  15  |
> > |        1       |   Aligned Code-centric   |   20  |  16 |  16 |   16  |  11  |  16  |
> > |        2       |   Aligned Summary+graph  |   14  |  13 |  15 |   16  |  15  |  15  |
> > |        2       |   Aligned Summary+code   |   22  |  24 |  22 |   15  |  17  |  20  |
> > |        3       |  Aligned graph   |   15  |  13 |  17 |   19  |  14  |  16  |
> > |        3       |   Aligned code   |   25  |  26 |  27 |   20  |  17  |  23  |
> > |        3       | Aligned summary  |   16  |  21 |  15 |   14  |  15  |  16  |
> > |        4       |    Only graph    |   16  |  28 |  19 |   16  |  14  |  19  |
> > |        4       |    Only code     |   25  |  26 |  27 |   20  |  17  |  23  |
> > |        4       |   Only summary   |   18  |  39 |  18 |   19  |  14  |  22  |
> > |                |  SOTA Baselines  |   17  |  16 |  25 |   26  |  16  |  20  |

---

> > > ### Comment · Reviewer_uzeD · 2024-11-25
> > > **Thanks.**
> > >
> > > Thanks for the response. I have increased my confidence. Good luck.

---

### Author Response · Authors · 2024-11-19
**General response to all reviewers**

We sincerely thank the four reviewers for their insightful and detailed evaluations of our manuscript. We appreciate that all reviewers have positively recognized multiple aspects of our work.

The novelty of our proposed multimodal fusion method for circuits is highly recognized by **Reviewer uzeD** *(this strategy is novel)*, **Reviewer kA13** *(overcomes the hardware unique (P1 to P4) looks novel and interesting)*, **Reviewer KB11** *(the first multimodal fused and implementation-aware circuit encoder)*.

The effectiveness and versatility of CircuitFusion in supporting multi-task evaluations have also been appreciated: **Reviewer uzeD** *(comprehensively evaluates multiple tasks)*, **Reviewer kA13** *(evaluations are comprehensive)*, **Reviewer bMDX** *(CircuitFusion outperforms SOTA task-specific supervised models in design quality evaluation task empirically)*, and **Reviewer bMDX** *(Sufficient ablation studies and experimental evaluations are conducted)*.


For the questions and concerns raised, we have carefully addressed them in our individual responses, hoping to resolve the concerns they may have. Additionally, we have updated the rebuttal version of our manuscript, with all modifications highlighted in blue for clarity. We thank the reviewers once again for their insightful comments, which have further improved our work.

---

### Author Response · Authors · 2024-11-25
**Official Comment by Authors**

Dear Reviewers,

Thank you for your valuable insights and constructive suggestions. We have made every effort to address your questions in our author response and are actively revising the paper in line with your feedback. As the discussion period is coming to a close, we were wondering if you could let us know whether our responses have addressed your concerns or if you have any additional questions. While we completely understand if the scores remain unchanged, we would greatly appreciate your feedback on our responses and are happy to provide further clarifications if needed.

Best regards,

Authors

---

### Meta-Review · Area_Chair_VakG · 2024-12-19

**Metareview:**

This paper presents CircuitFusion, a multimodal circuit encoder that integrates hardware code, structural graphs, and functionality summaries for agile chip design. The reviewers commended the paper's comprehensive evaluation and clear technical presentation. Initial concerns about technical overlap with RTLRewriter were addressed through detailed clarification of CircuitFusion's distinct focus on evaluation rather than generation tasks. Questions about the summary-centric fusion approach were resolved through new ablation studies demonstrating its superiority over alternative fusion strategies.
The authors have strengthened the work during review, particularly in demonstrating CircuitFusion's handling of multi-clock scenarios and explaining how their register-based splitting strategy expands limited RTL designs into a substantial training set. Moving forward, I suggest expanding the investigation of multi-clock domain handling and providing more detailed downstream application scenarios.

Given the paper's novel contribution to circuit representation learning and the authors' thorough engagement with reviewer feedback, I recommend acceptance.

**Additional Comments On Reviewer Discussion:**

The reviewers commended the paper's comprehensive evaluation and clear technical presentation. Initial concerns about technical overlap with RTLRewriter were addressed through detailed clarification of CircuitFusion's distinct focus on evaluation rather than generation tasks. Questions about the summary-centric fusion approach were resolved through new ablation studies demonstrating its superiority over alternative fusion strategies.
The authors have strengthened the work during review, particularly in demonstrating CircuitFusion's handling of multi-clock scenarios and explaining how their register-based splitting strategy expands limited RTL designs into a substantial training set.

---

### Decision · Program_Chairs · 2025-01-22

Accept (Poster)